# Asynchronous Federated Reinforcement Learning with Policy Gradient Updates: Algorithm Design and Convergence Analysis

**Guangchen Lan**[1], **Dong-Jun Han**[2], **Abolfazl Hashemi**[1], **Vaneet Aggarwal**[1],
**Christopher G. Brinton**[1]

[1] Purdue University, West Lafayette, IN, USA,  [2] Yonsei University, Seoul, South Korea

[1] {lan44,abolfazl,vaneet,cgb}@purdue.edu,  [2] djh@yonsei.ac.kr

## Abstract

To improve the efficiency of reinforcement learning (RL), we propose a novel asynchronous federated reinforcement learning (FedRL) framework termed AFedPG, which constructs a global model through collaboration among $N$ agents using policy gradient (PG) updates. To address the challenge of lagged policies in asynchronous settings, we design a delay-adaptive lookahead technique *specifically for FedRL* that can effectively handle heterogeneous arrival times of policy gradients. We analyze the theoretical global convergence bound of AFedPG, and characterize the advantage of the proposed algorithm in terms of both the sample complexity and time complexity. Specifically, our AFedPG method achieves $\mathcal{O}(\frac{\epsilon^{-2.5}}{N})$ sample complexity for global convergence at each agent on average. Compared to the single agent setting with $\mathcal{O}(\epsilon^{-2.5})$ sample complexity, it enjoys a linear speedup with respect to the number of agents. Moreover, compared to synchronous FedPG, AFedPG improves the time complexity from $\mathcal{O}(\frac{t_{\max}}{N})$ to $\mathcal{O}(\sum_{i=1}^{N} \frac{1}{t_i})^{-1}$, where $t_i$ denotes the time consumption in each iteration at agent $i$, and $t_{\max}$ is the largest one. The latter complexity $\mathcal{O}(\sum_{i=1}^{N} \frac{1}{t_i})^{-1}$ is always smaller than the former one, and this improvement becomes significant in large-scale federated settings with heterogeneous computing powers ($t_{\max} \gg t_{\min}$). Finally, we empirically verify the improved performance of AFedPG in four widely used MuJoCo environments with varying numbers of agents. We also demonstrate the advantages of AFedPG in various computing heterogeneity scenarios.

## 1 Introduction

Policy gradient (PG) methods, also known as REINFORCE (Williams, 1992), are widely used to solve reinforcement learning (RL) problems across a range of applications, with recent notable examples including reinforcement learning from human feedback (RLHF) in Google's Gemma (Team & DeepMind, 2024), OpenAI's InstructGPT (Ouyang et al., 2022), and ChatGPT (GPT-4 (OpenAI, 2023)). PG-based methods has also garnered significant attention in several applications including transportation systems (Al-Abbasi et al., 2019), networking (Geng et al., 2023), data-center resource allocation (Chen et al., 2023; 2024a; Zhang et al., 2023), streaming (Elgabli et al., 2024; Liu & Zhu, 2023), robotics (Gonzalez et al., 2023), and diffusion models (Zhang et al., 2024b).

Most practical RL applications operate at large scales and rely on a huge amount of data samples for model training in an online behavior (Provost & Fawcett, 2013; Liu et al., 2021), which is considered as a key bottleneck in RL (Dulac-Arnold et al., 2021; Ladosz et al., 2022). To reduce sample complexity while enabling training on massive data, a conventional approach involves transmitting locally collected samples from distributed agents to a central server (Predd et al., 2006). The transmitted samples can then be used for policy learning on the server side. However, this may not be feasible in real-world mobile systems where the communication bandwidth is limited and a large training time is intolerable (Brinton et al., 2025; Chu et al., 2024a; Yuan et al., 2024; Niknam et al., 2020; Elgabli et al., 2020), such as wireless edge devices (Lan et al., 2023a; Ching et al., 2023; 2024),

Table 1: Performance improvements of our AFedPG over other first-order policy gradient methods. We compare the complexity for convergence in each agent.

| Methods | Sample Complexity (FOSP) | Sample Complexity (Global) | Global Time |
|---|---|---|---|
| Vanilla PG (Yuan et al., 2022) | $\mathcal{O}(\epsilon^{-4})$ | $\mathcal{O}(\epsilon^{-3})$ | - |
| Normalized PG (Fatkhullin et al., 2023) | $\mathcal{O}(\epsilon^{-3.5})$ | $\mathcal{O}(\epsilon^{-2.5})$ | - |
| FedPG | $\mathcal{O}(\frac{\epsilon^{-3.5}}{N})$ | $\mathcal{O}(\frac{\epsilon^{-2.5}}{N})$ | $\mathcal{O}(\frac{t_{\max}}{N}\epsilon^{-2.5})$ |
| **AFedPG** | $\mathcal{O}(\frac{\epsilon^{-3.5}}{N})$ | $\mathcal{O}(\frac{\epsilon^{-2.5}}{N})$ | $\mathcal{O}(\bar{t}\epsilon^{-2.5})$ |

cloud computing (Chen et al., 2024b; Xu et al., 2024), autonomous driving (Kiran et al., 2022), and financial applications (Long et al., 2024; Liang et al., 2024). In RL, since new data is continuously generated based on the current policy, the samples collected at the agents need to be transmitted frequently to the server throughout the training process (Shen et al., 2023), resulting in a significant bottleneck with large delays. Furthermore, the sharing of individual data collected by agents may lead to privacy and legal issues (Kairouz et al., 2021; Mothukuri et al., 2021).

Federated learning (FL) (McMahan et al., 2017) offers a promising solution to the above challenges in a distributed setting. In FL, instead of directly transmitting the raw datasets to the central server, agents only communicate locally trained model parameters (or gradients) with the server. In Chu et al. (2025; 2024b), it shows that FL could achieve fairness among different agents. Although FL has been mostly studied for supervised learning problems, several recent works expanded the scope of FL to federated reinforcement learning (FedRL), where $N$ agents collaboratively learn a global policy without sharing the trajectories they collected during agent-environment interaction (Jin et al., 2022; Khodadadian et al., 2022; Xie & Song, 2023). FedRL has been studied in the tabular case (Agarwal et al., 2021b; Jin et al., 2022; Khodadadian et al., 2022), and for value-function based algorithms (Wang et al., 2024b; Xie & Song, 2023), where a linear speedup has been demonstrated. For policy-gradient based algorithms, namely FedPG, we note that a linear speedup is easy to achieve, given that trajectories collected at different agents can be processed in parallel (Lan et al., 2023b). Ganesh et al. (2024a) proposed a policy gradient-based approach that is robust to adversarial agents and achieves optimal sample complexity in the presence of adversaries. Liu & Zhu (2022) analyzes the performances in the multi-agent setting. Zhu et al. (2024) extends the result to the multi-task setting. However, policy-gradient based FedRL has not been well studied in terms of *global* (non-local) time complexity.

Despite all the aforementioned works in FedRL, they still face challenges in terms of time complexity, primarily due to their focus on synchronous model aggregation. In large-scale heterogeneous settings (Xiong et al., 2024; Xie et al., 2020; Chen et al., 2020), performing synchronous global updates has limitations, as the overall time consumption heavily depends on the slow agents, *i.e.*, stragglers (Badita et al., 2021; Mishchenko et al., 2022). In this paper, we aim to tackle this issue by strategically leveraging asynchronous federated learning (A-FL) in policy-based FedRL for the first time. A-FL (Xie et al., 2020; Chang et al., 2024) shows superiority compared to synchronous FL, and recent works (Mishchenko et al., 2022; Koloskova et al., 2022) further improve convergence performances with theoretical guarantees.

**Challenges:** However, compared to prior A-FL approaches focusing on supervised learning, integrating A-FL with policy-based FedRL introduces new challenges due to the presence of *lagged policies* in asynchronous settings. Unlike supervised FL where the datasets of the clients are fixed, in RL, agents collect new samples in each iteration based on the current policy. This dynamic nature of the data collection process makes both the problem itself and the theoretical analysis challenging. Guaranteeing the global convergence of the algorithm is especially non-trivial under the asynchronous FedRL setting with lagged policies. This problem setting and its challenges have been largely overlooked in existing research, despite the significance of employing FL in RL. The key question that this paper aims to address is:

*Despite the inherent challenge of dealing with lagged policies among different agents, can we improve the efficiency of FedPG through asynchronous methods while ensuring theoretical convergence?*

We answer this question in the affirmative by proposing AFedPG, an algorithm that asynchronously updates the global policy using policy gradients from federated agents. The key components of the proposed approach include a delay-adaptive lookahead technique tailored to PG, which addresses the inconsistent arrival times of updates during the training process. This new approach eliminates the second-order correction terms that do not appear in conventional supervised FL, effectively addressing the unique challenges of asynchronous FedRL. The improvement of AFedPG over other approaches is summarized in Table 1. Here, the global time is measured by the number of global updates $\times$ time complexity in each iteration. In terms of sample complexity, the federated learning technique brings a linear speedup with respect to the number of agents $N$. As for the global time, AFedPG improves from $\mathcal{O}(\frac{t_{\max}}{N}\epsilon^{-2.5})$ to $\mathcal{O}(\bar{t}\epsilon^{-2.5})$, where $\bar{t} := 1/\sum_{i=1}^{N}\frac{1}{t_i}$ is a harmonic average which is less than or equal to $t_{\max}/N$, and $t_i$ denotes the time complexity in each iteration at agent $i$.

### 1.1 SUMMARY OF CONTRIBUTIONS

Our main contributions can be summarized as follows:

1. **New methodology with a delay-adaptive technique:** We propose AFedPG, an asynchronous training method tailored to FedRL. To handle the delay issue in the asynchronous FedRL setting, we design a delay-adaptive lookahead technique. Specifically, in the $k$-th iteration of training, the agent collects samples according to the local model parameters $\widetilde{\theta}_k \leftarrow \theta_k + \frac{1-\alpha_{k-\delta_k}}{\alpha_{k-\delta_k}}(\theta_k - \theta_{k-1})$, where $\delta_k$ is the delay. Unlike (supervised) FL, a second-order correction term (marked blue in equation 32) **only** occurs in RL because of the sampling mechanism. This updating technique cancels out the second-order correction terms $((1-\alpha_{k-\delta_k})\nabla^2 J(\theta_k)(\theta_{k-1}-\theta_k) + \alpha_{k-\delta_k}\nabla^2 J(\theta_k)(\widetilde{\theta}_k - \theta_k) = 0)$ and thus assists the convergence analysis. This technique is specifically designed for AFedRL, and not developed by previous FL works.

2. **Convergence analysis:** This work gives both the *global* and the first-order stationary point (FOSP) convergence guarantees of the asynchronous federated policy-based RL for the **first time**. We analytically characterize the convergence bound of AFedPG using the key lemmas, and show the impact of various parameters including delay and number of iterations.

3. **Linear speedup in sample complexity:** As shown in Table 1, our AFedPG approach improves the sample complexity in each agent from $\mathcal{O}(\epsilon^{-2.5})$ (single agent PG) to $\mathcal{O}(\frac{\epsilon^{-2.5}}{N})$, where $N$ is the number of federated agents. This represents the linear speedup of our method with respect to the number of agents $N$.

4. **Time complexity improvement:** Our AFedPG also reduces the time complexity of synchronous FedPG from $\mathcal{O}(\frac{t_{\max}}{N})$ to $\mathcal{O}(\bar{t} := \frac{1}{\sum_{i=1}^{N}\frac{1}{t_i}})$. The latter is always smaller than the former. This improvement is significant in large-scale federated settings with heterogeneous delays ($t_{\max} \gg t_{\min}$).

5. **Experiments under the MuJoCo environment:** We empirically verify the improved performances of AFedPG in four different MuJoCo environments with varying numbers of agents. We also demonstrate the improvements with different computing heterogeneity.

To the best of our knowledge, this is the first work to successfully integrate policy-based reinforcement learning with asynchronous federated learning and analyze its behavior, accompanied by theoretical convergence guarantees. This new setting necessitates us to deal with the lagged policies under a time-varying data scenario depending on the updated policy.

**Notation:** We denote the Euclidean norm by $\|\cdot\|$, and the vector inner product by $\langle\cdot\rangle$. For a vector $a \in \mathbb{R}^n$, we use $a^\top$ to denote the transpose of $a$. A calligraphic font letter denotes a set, *e.g.*, $\mathcal{C}$, and $|\mathcal{C}|$ denotes its cardinality. We use $\mathcal{C} \setminus \{j\}$ to denote a set that contains all the elements in $\mathcal{C}$ except for $j$.

## 2 RELATED WORK

In this section, we review previous works that are most relevant to the present work.

**Policy gradient methods:** For vanilla PG, the state-of-the-art result is presented in Yuan et al. (2022), achieving a sample complexity of $\widetilde{\mathcal{O}}(\epsilon^{-4})$ for the local convergence. Several recent works have improved this boundary with PG variants. In Huang et al. (2020), a PG with momentum method is proposed with convergence rate $\mathcal{O}(\epsilon^{-3})$ for the local convergence. The authors of Ding et al. (2022) further improve the convergence analysis of PG with momentum and achieve a global convergence with the rate $\mathcal{O}(\epsilon^{-3})$. In Fatkhullin et al. (2023), a normalized PG technique is introduced and improves the global convergence rate to $\mathcal{O}(\epsilon^{-2.5})$. In Mondal & Aggarwal (2024), an acceleration-based natural policy gradient method is proposed with sample complexity $\mathcal{O}(\epsilon^{-2})$, but second-order matrices are computed in each iteration (with first-order information), which brings more computational cost. However, all previous works have focused on the single-agent scenario, and the federated PG has not been explored. Compared to these works, we focus on a practical federated PG setting with distributed agents to improve the efficiency of RL. Our scheme achieves a linear speedup with respect to the number of agents, significantly reducing the sample complexity compared to the conventional PG approaches.

**Asynchronous FL:** In Xie et al. (2020), the superiority of asynchronous federated learning (A-FL) has been empirically shown compared to synchronous FL in terms of convergence performances. In Chen et al. (2020), the asynchronous analysis is extended to the vertical FL with a better convergence performance compared to the synchronous setting. In Dun et al. (2023), a dropout regularization method is introduced to handle the heterogeneous problems in A-FL. About the same time, recent works (Mishchenko et al., 2022; Koloskova et al., 2022) further improve the convergence performance with theoretical guarantees, and show that A-FL always exceeds synchronous FL without any changes to the algorithm. We note that all previous A-FL works focus on supervised learning with fixed datasets on the client side. However, in RL, agents collect new samples that depend on the current policy (model parameters) in each iteration, which makes the problem fundamentally different and challenging. In this work, we address this challenge by developing an A-FL method highly tailored to policy gradient, leveraging the proposed delay-adaptive lookahead technique.

**FedRL:** For value-function based algorithms, Zheng et al. (2024); Jin et al. (2022); Woo et al. (2023) analyze the convergence performances with environment heterogeneity. Salgia & Chi (2024) analyzes the trade-off between sample and communication complexity. Zheng et al. (2024); Khodadadian et al. (2022) shows a linear speedup with respect to the number of agents. Zhang et al. (2024a) extends the result with linear function approximation. However, all of the above works are limited to tabular or linear approximation analysis (without deep learning). For actor-critic (AC) based method, Wang et al. (2024b) analyzes the convergence performances with the linear function approximation. Mnih et al. (2016) builds a practical system to implement the neural network approximation. Yang et al. (2024) analyzes the sample complexity in the multi-task setting. In Xie & Song (2023), it adds KL divergence and experimentally validates the actor-critic based method with neural network approximation. For policy-based methods, Chen et al. (2021) gives a convergence guarantee for the vanilla FedPG. (Wang et al., 2024a) analyzes the performance in the heterogeneous setting. Lan et al. (2023b) further shows the simplicity compared to the other RL methods, and a linear speedup has been demonstrated in the synchronous setting. Further, optimal sample complexity for global optimality in federated RL even in the presence of adversaries is studied in Ganesh et al. (2024a). However, with online behavior, it is not practical to perform synchronous global updates with heterogeneous computing power, and the global time consumption heavily depends on the stragglers (Mishchenko et al., 2022). This motivates us to consider the asynchronous policy-based method for FedRL. We demonstrate both theoretically and empirically that our method further reduces the time complexity compared to the synchronous FedRL approach.

## 3 PROBLEM SETUP

**Markov decision process:** We consider the Markov decision process (MDP) as a tuple $(\mathcal{S}, \mathcal{A}, \mathcal{P}, \mathcal{R}, \gamma)$, where $\mathcal{S}$ is the state space, $\mathcal{A}$ is a finite action space, $\mathcal{P} : \mathcal{S} \times \mathcal{A} \times \mathcal{S} \to \mathbb{R}$ is a Markov kernel that determines transition probabilities, $\mathcal{R} : \mathcal{S} \times \mathcal{A} \to \mathbb{R}$ is a reward function, and $\gamma \in (0, 1)$ is a discount factor. At each time step $t$, the agent executes an action $a_t \in \mathcal{A}$ from the

current state $s_t \in \mathcal{S}$, following a stochastic policy $\pi$, *i.e.*, $a_t \sim \pi(\cdot|s_t)$. The corresponding reward is defined as $r_t$. The state value function is defined as

$$V_\pi(s) = \mathop{\mathbb{E}}_{\substack{a_t \sim \pi(\cdot|s_t), \\ s_{t+1} \sim P(\cdot|s_t, a_t)}} \left[ \sum_{t=0}^{\infty} \gamma^t r(s_t, a_t) | s_0 = s \right]. \tag{1}$$

Similarly, the state-action value function (Q-function) is defined as

$$Q_\pi(s, a) = \mathop{\mathbb{E}}_{\substack{a_t \sim \pi(\cdot|s_t), \\ s_{t+1} \sim P(\cdot|s_t, a_t)}} \left[ \sum_{t=0}^{\infty} \gamma^t r(s_t, a_t) | s_0 = s, \ a_0 = a \right]. \tag{2}$$

An advantage function is then define as $A_\pi(s, a) = Q_\pi(s, a) - V_\pi(s)$. With continuous states, the policy is parameterized by $\theta \in \mathbb{R}^d$, and then the policy is referred as $\pi_\theta$ (Deep RL parameterizes $\pi_\theta$ by deep neural networks). A state-action visitation measure induced by $\pi_\theta$ is given as

$$\nu_{\pi_\theta}(s, a) = (1 - \gamma) \mathop{\mathbb{E}}_{s_0 \sim \rho} \left[ \sum_{t=0}^{\infty} \gamma^t P(s_t = s, \ a_t = a | s_0, \ \pi_\theta) \right], \tag{3}$$

where the starting state $s_0$ is drawn from a distribution $\rho$. The goal of the agent is to maximize the expected discounted return defined as follows:

$$\max_\theta J(\theta) \coloneqq \mathop{\mathbb{E}}_{s \sim \rho} \left[ V_{\pi_\theta}(s) \right]. \tag{4}$$

The gradient of $J(\theta)$ (Schulman et al., 2018) can be written as:

$$\nabla_\theta J(\theta) = \mathop{\mathbb{E}}_\tau \left[ \sum_{t=0}^{\infty} \nabla_\theta \log \pi_\theta(a_t|s_t) \cdot A_{\pi_\theta}(s_t, a_t) \right], \tag{5}$$

where $\tau = (s_0, a_0, r_0, s_1, a_1, r_1 \cdots)$ is a trajectory induced by policy $\pi_\theta$. We omit the $\theta$ notation in the gradient operation and denote the policy gradient by $g$ for short. Then, $g$ is estimated by

$$g(\theta, \tau) = \sum_{t=0}^{\infty} \nabla \log \pi_\theta(a_t|s_t) \sum_{h=t}^{\infty} \gamma^h r(s_h, a_h). \tag{6}$$

**Federated policy gradient:** We aim to solve the above problem in an FL setting, where $N$ agents collaboratively train a common policy $\pi_\theta$. Specifically, each agent collects trajectories and corresponding reward $r(s_h, a_h)$ based on its local policy. Then, each agent $i$ estimates $g(\theta_i, \tau)$ for training the model $\theta$, and the updated models are aggregated at the server. Motivated by the limitations of synchronous model aggregation in terms of time complexity, in the next section, we present our AFedPG methodology that takes an asynchronous approach to solve equation 4 in an FL setting.

## 4 PROPOSED ASYNCHRONOUS FEDPG

The proposed algorithm consists of $K$ global iterations, indexed by $k = 0, 1, \ldots, K - 1$. We first introduce the definition of concurrency and delay in our asynchronous federated setting. We then present the proposed AFedPG methodology.

**Definition 4.1.** (Concurrency) We denote $\mathcal{C}_k$ as the set of active agents in the $k$-th global iteration, and define $\omega_k \coloneqq |\mathcal{C}_k|$ as the concurrency. We define the average and maximum concurrency as $\bar{\omega} \coloneqq \frac{1}{K} \sum_{k=0}^{K-1} \omega_k$, and $\omega_{\max} \coloneqq \max \omega_k$, respectively.

In each global iteration of AFedPG, the server applies only one gradient to update the model from the agent who has finished its local computation, while the other $N - 1$ agents keep computing local gradients (unapplied gradients) in parallel. In this asynchronous setting, the models used in each agent are outdated, as the server keeps updating the model. Thus, we introduce the notion of delay (or staleness) $\delta \in \mathbb{N}$, which measures the difference between the current global iteration and the past global iteration when the agent received the updated model from the server.

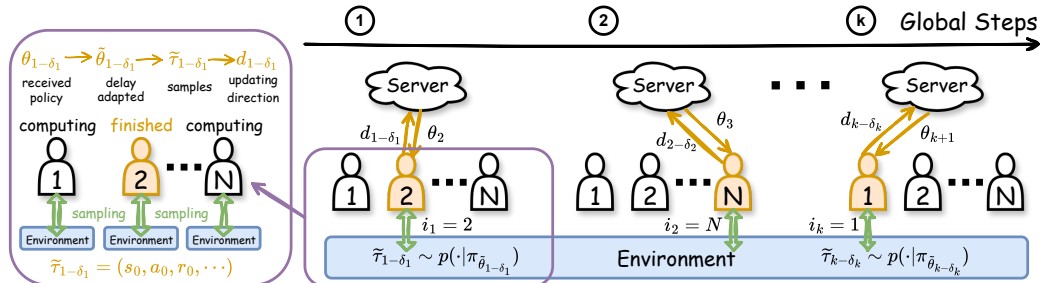

Figure 1: An illustration of the asynchronous federated policy gradient updates. Each agent has a local copy of the environment, and agents may collect data according to different local policies. At each iteration, the agent in the yellow color finishes the local process and then communicates with the server, while the other agents keep sampling and computing local gradients in parallel. In the $k$-th global iteration, $\delta_k \in \mathbb{N}$ is the delay, $\widetilde{\tau}_{k-\delta_k}$ is the sample collected according to the policy $\pi_{\widetilde{\theta}_{k-\delta_k}}$, and $d_{k-\delta_k}$ is the updating direction calculated from the sample $\widetilde{\tau}_{k-\delta_k}$.

**Definition 4.2.** (Delay) In the $k$-th global iteration, we denote the delay of the applied gradient as $\delta_k \in \mathbb{N}$ (an integer), and the delays of the unapplied gradients as $\{\delta_k^i\}_{i \in \mathcal{C}_k \setminus \{j_k\}}$, where $j_k$ denotes the agent that communicates with the server. $\delta_k^i$ is the difference between the $k$-th iteration and the iteration where the agent $i$ started to compute the latest gradient. In the final $K$-th global iteration, we have $K$ applied gradients $\{\delta_k\}_{k=0}^{K-1}$ and unapplied gradients $\{\delta_k^i\}_{i \in \mathcal{C}_K \setminus \{j_K\}}$. The average delay can be expressed as

$$\overline{\delta} := \frac{1}{K-1+|\mathcal{C}_K|}\Big(\sum_{k=0}^{K-1} \delta_k + \sum_{i \in \mathcal{C}_K \setminus \{j_K\}} \delta_K^i\Big). \tag{7}$$

**Asynchronous FedPG:** Our goal is to train a global policy with parameters $\theta \in \mathbb{R}^d$ via FL across $N$ distributed agents. As shown in Figure 1, during the training process, agent $i$ collects trajectories in an online behavior, and computes gradients or updating directions using its *local trajectories* (also known as samples). Then, agent $i$ transmits local gradients to the central server. In particular, in the $k$-th global iteration, the training of our AFedPG consists of the following three steps:

- **Local computation and uplink transmission:** Agent $i$ receives the previous policy $\pi_{\theta_j}$ from the server in the $j$-th global iteration. Agent $i$ collects its own local trajectory $\tau_j$ based on its current policy $\pi_{\theta_j}$. Agent $i$ then computes its local updating direction $d_j \in \mathbb{R}^d$ based on the trajectory $\tau_j$, and sends it back to the server.

- **Server-side model update:** The server starts to operate the $k$-th global iteration as soon as it receives $d_j$ from agent $i$. Thus, denote $d_{k-\delta_k} = d_j$, where $\delta_k$ is the delay in the $k$-th global iteration. The server updates global policy parameters by $\theta \leftarrow \theta - \eta d_{k-\delta_k}$, where $\eta$ is the learning rate.

- **Downlink transmission:** The server transmits the current global policy parameters $\theta \in \mathbb{R}^d$ back to the agent $i$ as soon as it finishes the global update.

The server side procedure of AFedPG is shown in Algorithm 1 and the process at the agent side is shown in Algorithm 2. In the $k$-th global iteration, the server operates one global update (Steps 4 and 5) as soon as it receives a direction $d_{k-\delta_k}$ from an agent with delay $\delta_k$. After the global update, the server sends the updated model back to that agent. In Algorithm 2, after receiving the global model from the server, an agent first gets model parameters $\widetilde{\theta}_k$ according to Step 2, and then collects samples based on the policy $\pi_{\widetilde{\theta}_k}$ (Steps 3). At last, the agent computes the updating direction $d_k$, and sends it to the server as soon as it finishes the local process. Overall, all agents conduct local computation in parallel, but the global model is updated in an asynchronous manner as summarized in Figure 1.

**Normalized update at the server:** In Step 5 of Algorithm 1, to handle the updates with various delays, we use normalized gradients with controllable sizes. Specifically, the error term $\|e_k\|$ in

---

**Algorithm 1** AFedPG: Server.

---

**Require:** MDP $(\mathcal{S}, \mathcal{A}, \mathcal{P}, \mathcal{R}, \gamma)$; Number of iterations $K$; Step size $\eta_k, \alpha_k$; Initial $\theta_0, d_0 \in \mathbb{R}^d$.
1: Broadcast $\theta_0$ to $N$ agents.
2: **for** $k = 1, \cdots, K$ **do**
3:     ▷ Uplink Transmit
4:     Receive $g(\widetilde{\tau}_{k-\delta_k}, \widetilde{\theta}_{k-\delta_k})$ from the agent $i_k$.
5:     $d_{k-\delta_k} \leftarrow (1 - \alpha_{k-\delta_k})d_{k-1-\delta_{k-1}} + \alpha_{k-\delta_k}g(\widetilde{\tau}_{k-\delta_k}, \widetilde{\theta}_{k-\delta_k})$
6:     ▷ Server update
7:     $\theta_k \leftarrow \theta_{k-1} + \eta_{k-1}\frac{d_{k-\delta_k}}{\|d_{k-\delta_k}\|}$
8:     $\widetilde{\theta}_k \leftarrow \theta_k + \frac{1-\alpha_{k-\delta_k}}{\alpha_{k-\delta_k}}(\theta_k - \theta_{k-1})$    # Lookahead
9:     ▷ Downlink Transmit
10:     Transmit $\widetilde{\theta}_k$ back to the agent $i_k$.
11: **end for**
**Ensure:** $\theta^K$

---

**Algorithm 2** AFedPG: Agent $i$ Update, $i = 1, \cdots, N$.

---

**Require:** $\widetilde{\theta}_{k'} \in \mathbb{R}^d$
1: Receive $\widetilde{\theta}_{k'}$ from the server.
2: $\widetilde{\tau}_{k'} \sim p(\cdot|\pi_{\widetilde{\theta}_{k'}})$   # Sampling
3: Estimate policy gradient $g(\widetilde{\tau}_{k'}, \widetilde{\theta}_{k'})$ according to equation 6.
4: When finish computing, transmit $g(\widetilde{\tau}_{k'}, \widetilde{\theta}_{k'})$ to the server.   # When the server receives the policy gradient, it is the $k$-th step on the server, where $k - \delta_k = k'$.
**Ensure:** $g(\widetilde{\tau}_{k'}, \widetilde{\theta}_{k'})$

---

Lemma B.9 is related to $\|\nabla J(\widetilde{\theta}_{k-\delta_k}) - \nabla J(\widetilde{\theta}_k)\|$ and $\|\nabla J(\theta_{k-1}) - \nabla J(\theta_k)\|$. With the smoothness in Lemma B.7, we are able to bound the error as $\|\theta_{k-1} - \theta_k\| = \eta_{k-1}$. The details of the boundaries are shown in equation 34.

**Delay-adaptive lookahead at the agent:** In Step 7 of Algorithm 1 (in blue), we design a delay-adaptive lookahead update technique, which is designed specifically for asynchronous FedPG. It operates as follows:

$$\theta_k = (1 - \alpha_{k-\delta_k})\theta_{k-1} + \alpha_{k-\delta_k}\widetilde{\theta}_k. \tag{8}$$

We note that equation 8 is not the conventional momentum method, because the orders are different. Here, $\widetilde{\theta}_k$ "looks ahead" based on global policies $\theta_{k-1}$ and $\theta_k$, and the learning rate $\alpha_{k-\delta_k}$ depends on the delay $\delta_k$.

In equation 8, the agent collects samples according to the current parameter $\widetilde{\theta}_k$ in an online behavior, which only happens in the RL setting. This makes the problem fundamentally different from the conventional FL on supervised learning with a fixed dataset. This mechanism cancels out the second-order Hessian correction terms:

$$(1 - \alpha_{k-\delta_k})\nabla^2 J(\theta_k)(\theta_{k-1} - \theta_k) + \alpha_{k-\delta_k}\nabla^2 J(\theta_k)(\widetilde{\theta}_k - \theta_k) \to 0, \tag{9}$$

and thus assists the convergence analysis. The details of the derivation are shown in Appendix B.4 marked in blue.

## 5   CONVERGENCE ANALYSIS

In this section, we derive the convergence rates of AFedPG with the following criterions for the FOSP and global convergence, respectively.

**Global Criterion:** We focus on the global convergence, *i.e.*, finding the parameter $\theta$ *s.t.* $J^\star - J(\theta) \leq \epsilon'$, where $J^\star$ is the optimal expected return.

**FOSP Criterion:** We focus on the first-order stationary convergence, *i.e.*, finding the parameter $\theta$ *s.t.* $\|\nabla J(\theta)\| \leq \epsilon$.

We use several standard assumptions listed in Appendix B.1. Based on these assumptions, the convergence rates of AFedPG are given in Theorem 5.2 and 5.1.

**Theorem 5.1.** *(Global) Let Assumption B.1 and B.2 hold. With suitable learning rates $\eta_k$ and $\alpha_k$, after $K$ global iterations, AFedPG satisfies*

$$J^\star - \mathbb{E}[J(\theta_K)] \leq \mathcal{O}\big(K^{-\frac{2}{5}} \cdot (1-\gamma)^{-3}\big) + \frac{\sqrt{\epsilon_{\text{bias}}}}{1-\gamma}, \tag{10}$$

*where $\epsilon_{\text{bias}}$ is from equation 13. Thus, to satisfy $J^\star - J(\theta_K) \leq \epsilon + \frac{\sqrt{\epsilon_{\text{bias}}}}{1-\gamma}$, we need $K = \mathcal{O}(\frac{\epsilon^{-2.5}}{(1-\gamma)^{7.5}})$ iterations. As only one trajectory is required in each iteration, the number of trajectories is equal to $K$, i.e., the sample complexity is $\mathcal{O}(\frac{\epsilon^{-2.5}}{(1-\gamma)^{7.5}})$.*

**Theorem 5.2.** *(FOSP) Let Assumption B.1 hold. With suitable learning rates $\eta_k$ and $\alpha_k$, after $K$ global iterations, AFedPG satisfies*

$$\mathbb{E}[\|\nabla J(\bar{\theta}_K)\|] \leq \mathcal{O}\big(K^{-\frac{2}{7}} \cdot (1-\gamma)^{-3}\big), \tag{11}$$

*where $\mathbb{E}\|\nabla J(\bar{\theta}_K)\| := \frac{\sum_{k=1}^{K} \eta_k \mathbb{E}\|\nabla J(\theta_k)\|}{\sum_{k=1}^{K} \eta_k}$ is the average of gradient expectations. To satisfy $\nabla J(\bar{\theta}_K) \leq \epsilon$, we need $K = \mathcal{O}(\frac{\epsilon^{-3.5}}{(1-\gamma)^{7.5}})$ iterations. As only one trajectory is required in each iteration, the number of collected trajectories is equal to $K$, i.e., the sample complexity is $\mathcal{O}(\frac{\epsilon^{-3.5}}{(1-\gamma)^{7.5}})$.*

**Comparison to the synchronous setting:** Synchronous FedPG needs $\mathcal{O}(\epsilon^{-2.5})$ trajectories in total, and each agent needs $\mathcal{O}(\frac{\epsilon^{-2.5}}{N})$ trajectories. However, in synchronous FedGP the server has to wait for the slowest agent at each global step, which can still slow down the training process. Let $t_i$ denote the time consumption for agent $i = 1, \cdots, N$ at local steps with finite values. As the agent has the same computation requirement in each iteration (The number of collected samples is the same.), we assume that the time complexity in each iteration is the same. Then, the waiting time on the server for each step becomes $t_{\max} := \max t_i$ for FedPG. Our AFedPG approach keeps the same sample complexity as FedPG, but the server processes the global step as soon as it receives an update, speeding up training. Specifically, the average waiting time on the server is $\bar{t} := \frac{1}{\sum_{i=1}^{N} \frac{1}{t_i}} < \frac{t_{\max}}{N}$

at each step. Thus, the asynchronous FedPG achieves less time complexity than the synchronous approach regardless of the delay pattern. The advantage is significant when $t_{\max} \gg t_{min}$, which occurs in many practical settings with heterogeneous computation powers across different agents. We illustrate the advantage of AFedPG over synchronous FedPG in Figure 5 in Appendix A.1. As the server only operates one simple summation, without loss of generality, the time consumption at the server side is negligible.

## 6 Experiments

### 6.1 Setup

**Environment:** To validate the effectiveness of our approach via experiments, we consider four popular MuJoCo environments for robotic control (Swimmer-v4, Hopper-v4, Walker2D-v4, and Humanoid-v4) (Todorov et al., 2012) with the MIT License. Both the state and action spaces are continuous. Environmental details are described in Table 2 in Appendix A.2, and the MuJoCo tasks are visualized in Figure 2.

**Measurement:** All convergence performances are measured over 10 runs with random seeds from 0 to 9. The solid lines in our main experimental results are the averaged results, and the shadowed areas are confidence intervals with the confidence level 95%. The lines are smoothed for better visualization.

**Implementation:** Policies are parameterized by fully connected multi-layer perceptions (MLPs) with settings listed in Table 3 in Appendix A.1. We follow the practical settings in stable-baselines3 (Raffin et al., 2021) to update models with generalized advantage estimation (GAE) (0.95) (Schulman et al., 2018) in our implementation. We use PyTorch (Paszke et al., 2019) to implement deep neural networks (DNNs). All tasks are trained on NVIDIA A100 GPUs with 40 GB of memory.

**Baselines:** We first consider the conventional PG approach with $N = 1$, to see the effect of using multiple agents for improving sample complexity. We then consider the synchronous FedPG method as a baseline to observe the impact of asynchronous updates on enhancing the time complexity. To see the effect of our delay-adaptive technique, we also consider the performance of AFedPG without the delay-adaptive updates, namely vanilla in Figure 4. Finally, we consider A3C (Mnih et al., 2016), an asynchronous method designed for RL. We note that only a few prior works could be used on the federated PG problem, *e.g.*, A3C, and many existing works in federated supervised learning are not directly applicable to our federated RL setting.

**Performance metrics**: We consider the following metrics:

1. Rewards: the average trajectory rewards collected at each iteration;
2. Convergence: rewards versus iterations during the training process;
3. Time consumption: global time with certain numbers of collected samples.

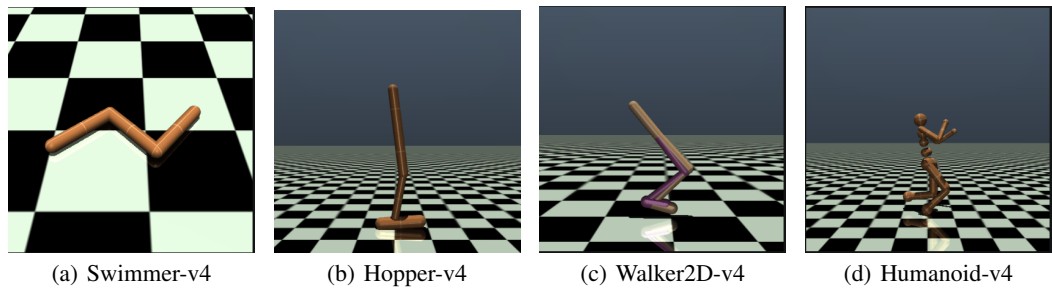

(a) Swimmer-v4      (b) Hopper-v4      (c) Walker2D-v4      (d) Humanoid-v4

Figure 2: Visualization of the four MuJoCo tasks considered in this paper for experiments.

## 6.2 RESULTS

**Sample complexity improvement:** First, to verify the improvement of sample complexity in the first row of Table 1, we evaluate the speedup effects of the number of federated agents $N$. In Figure 3, with different numbers of agents, we test the convergence performances of AFedPG ($N = 2, 4, 8$) and the single agent PG ($N = 1$). The x-axis is the number of samples collected by each agent on average, and the y-axis is the reward. In all four MuJoCo tasks, AFedPG beats the single-agent PG: AFedPG converges faster, has lower variances, and achieves higher final rewards when more agents are involved in collecting trajectories and estimating policy gradients. These results confirm the advantage of AFedPG in terms of sample complexity.

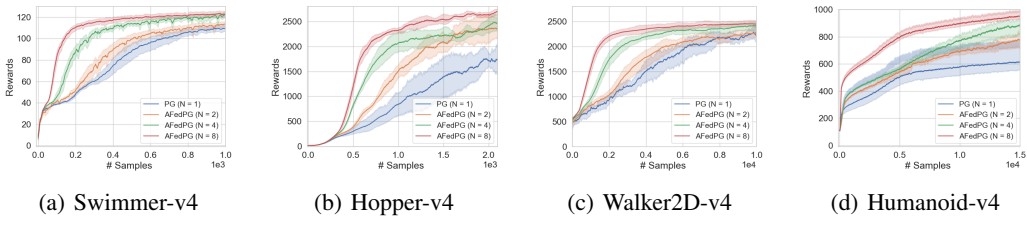

(a) Swimmer-v4      (b) Hopper-v4      (c) Walker2D-v4      (d) Humanoid-v4

Figure 3: Reward performances of AFedPG ($N = 2, 4, 8$) and PG ($N = 1$) on various MuJoCo environments, where $N$ is the number of federated agents. The solid lines are averaged results over 10 runs with random seeds from 0 to 9. The shadowed areas are confidence intervals with 95% confidence level.

**Speedup in global time complexity:** Second, to verify the improvement of global time complexity in the second row of Table 1, we compared the time consumption in the asynchronous and synchronous settings. In Figure 4, we set $N = 4, 8$ and fix the number of samples collected by all agents. Here, $t_{\max}$ is about 4 times more than $t_{\min}$. The numbers of total samples (trajectories) are $8 \times 10^3$,

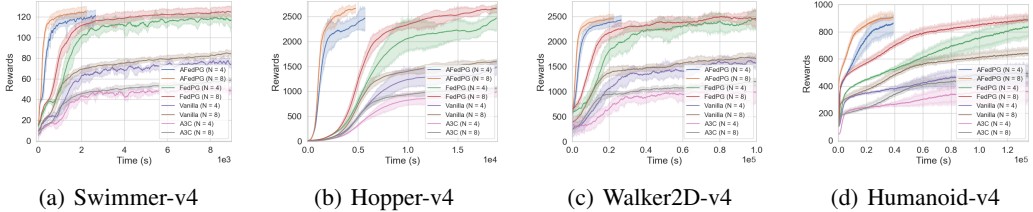

| (a) Swimmer-v4 | (b) Hopper-v4 | (c) Walker2D-v4 | (d) Humanoid-v4 |

Figure 4: Global time of AFedPG and FedPG with certain numbers of collected samples on various MuJoCo environments, where $N$ is the number of federated agents. The solid lines are averaged results over 10 runs. The shadowed areas are confidence intervals with $95\%$ confidence level.

$1.6 \times 10^4$, $8 \times 10^4$, and $1.2 \times 10^5$ for the Swimmer-v4 task, the Hopper-v4 task, the Walker2D-v4 task, and the Humanoid-v4 task individually when $N = 8$. When $N = 4$, the numbers are halved. In all four environments, AFedPG has much lower time consumption compared to the synchronous FedPG, confirming the enhancement in terms of time complexity. Compared to the A3C baseline, AFedPG achieves much higher rewards with less variance.

**Ablation study:** In Figure 4, we also observe the impact of our delay-adaptive lookahead approach. It is seen that the vanilla scheme without the delay-adaptive lookahead technique does not provide a satisfactory performance, confirming the importance of the proposed approach. We also analyze the effect of computation heterogeneity in Appendix A.3.

## 7 DISCUSSIONS

**Conclusions:** We proposed AFedPG, a novel asynchronous FedRL framework that updates the global model using PGs from multiple agents. To handle the challenge of lagged (heterogeneous) policies in the asynchronous setting, we designed a delay-adaptive lookahead technique and used normalized updates to integrate PGs. We then analytically characterized the convergence bound of AFedPG and showed both global and first-order stationary point convergence guarantees. We also showed that AFedPG achieves a speedup for both sample complexity and time complexity. First, our AFedPG method achieves $\mathcal{O}(\frac{\epsilon^{-2.5}}{N})$ sample complexity at each agent for global convergence. Compared to the SOTA result in the single agent setting, *i.e.* PG, with $\mathcal{O}(\epsilon^{-2.5})$ sample complexity, it enjoys a linear speedup with respect to the number of agents $N$. Second, compared to synchronous FedPG, AFedPG improves the time complexity from $\mathcal{O}(\frac{t_{\max}}{N})$ to $\mathcal{O}(\sum_{i=1}^{N} \frac{1}{t_i})^{-1}$, where $t_i$ denotes the time complexity in each iteration at the agent $i$, and $t_{\max}$ is the largest one. The latter complexity $\mathcal{O}(\sum_{i=1}^{N} \frac{1}{t_i})^{-1}$ is always smaller than the former one, and this improvement is significant in a large-scale federated setting with heterogeneous computing powers ($t_{\max} \gg t_{\min}$). Finally, we empirically verified the performances of AFedPG compared to various baselines in four MuJoCo environments with different $N$. We also demonstrated improvements with different computing heterogeneity. It is shown that AFedPG achieves speedup in terms of both sample complexity and time complexity, especially in scenarios with high computing power heterogeneity.

**Limitations and future work directions:** (i) Although the system is resilient to stragglers, addressing threats posed by adversarial or malicious workers in this setup remains an open problem. While robustness to such attacks has been explored in the synchronous setting (Ganesh et al., 2024a), extending these methods to an asynchronous environment is both a challenging and promising avenue for future research. (ii) It is worth studying whether the asynchronous method is compatible with other methods, *e.g.*, local update, quantization, and low-rank decomposition, to improve communication efficiency in the deep RL setting. (iii) Extending second-order policy optimization methods, such as natural policy gradient methods, which achieve optimal sample complexity in centralized settings (Mondal & Aggarwal, 2024), to an asynchronous federated setup is an important direction for future work. (iv) While this paper focuses on discounted rewards, exploring such results for average reward setup (Bai et al., 2024; Ganesh et al., 2025; 2024b) is an important open direction.

## REPRODUCIBILITY STATEMENT

In this statement, we discuss the efforts that have been made to ensure reproducibility.

For theoretical results, we clearly explain all assumptions in Appendix B.1, and a complete proof of the lemmas and theorems in Appendix B.

For algorithms, we provide the pseudocode in Algorithm 1 and Algorithm 2.

For datasets used in the experiments, we use open source datasets, and describe them in Section 6.1 and Appendix A.2.

## ACKNOWLEDGMENTS

This work was supported in part by the National Science Foundation (NSF) under grants CPS-2313109 and ITE-2326898, and by the Office of Naval Research (ONR) under grants N00014-22-1-2305 and N00014-23-1-2532.

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

# A  SUPPLEMENTARY RESULTS

In this section, we first compare the time complexity between the synchronous and the asynchronous settings. We then list the experimental settings in Appendix A.2. At last, we have supplementary experiments in Appendix A.3.

## A.1  COMPARISON TO THE SYNCHRONOUS SETTING

Let $T$ be the computation time during the entire training process to achieve a given number $K$ of cumulative communication rounds of all agents, where $K = \mathcal{O}(\epsilon^{-2.5})$ for a global convergence according to Theorem 5.1.

In AFedPG, for agent $i$, the number of communication rounds is $\frac{T}{t_i}$. For all agents, the total number is $\sum_{i=1}^{N} \frac{T}{t_i}$. As $\sum_{i=1}^{N} \frac{T}{t_i} = K$, we have $T = \frac{K}{\sum_{i=1}^{N} \frac{1}{t_i}} = \mathcal{O}(\frac{1}{\sum_{i=1}^{N} \frac{1}{t_i}} \epsilon^{-2.5})$ as a harmonic average of all agents.

In FedPG, the number of global communication rounds on the server is $\frac{T}{t_{\max}}$. As $\frac{T}{t_{\max}} = \frac{K}{N}$, we have $T = \frac{K t_{\max}}{N} = \mathcal{O}(\frac{t_{\max}}{N} \epsilon^{-2.5}) \geq \mathcal{O}(\frac{1}{\sum_{i=1}^{N} \frac{1}{t_i}} \epsilon^{-2.5})$ as the harmonic mean is always smaller or equal to the maximum one.

The asynchronous FedPG achieves better time complexity than the synchronous approach regardless of the delay pattern. The advantage is significant when $t_{\max} \gg t_{min}$, which occurs in many practical settings with heterogeneous computation powers across different agents. We illustrate the advantage of AFedPG over synchronous FedPG in Figure 5. As the server only operates one simple summation, without loss of generality, the time consumption at the server-side is negligible.

It is noticeable that we do not make any assumptions or requirements on $t_{\max}$ in the analysis of AFedPG. In the extreme case, the slowest agent does not communicate with the server, and thus, $t_{\max}$ is infinite. In this scenario, the time consumption of AFedPG does not hurt a lot, while the time consumption of FedPG becomes infinite.

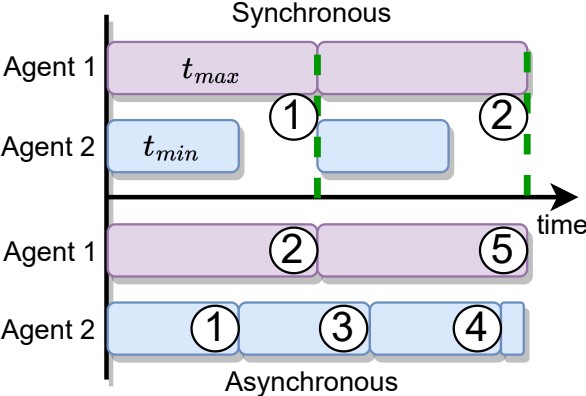

Figure 5: Comparison of time consumptions between synchronous and asynchronous approaches. The circled numbers denote the indices of global steps.

## A.2  SUPPLEMENTARY EXPERIMENTAL SETTINGS

In Section 6.1, we list the key experimental settings. We list the rest with details in this subsection. The environmental details (the four MuJoCo tasks) are described in Table 2. The policies $\pi_\theta$ are parameterized by fully connected multi-layer perceptions (MLPs) with settings listed in Table 3.

Table 2: Detailed descriptions of the four tasks in the MuJoCo environment.

| MuJoCo Tasks | Agent Types | Action Space Dimension | State Space Dimension |
|---|---|---|---|
| Swimmer-v4 | Three-link swimming robot | 2 | 8 |
| Hopper-v4 | Two-dimensional one-legged robot | 3 | 11 |
| Walker2D-v4 | Two-dimensional bipedal robot | 6 | 17 |
| Humanoid-v4 | Three-dimensional bipedal robot | 17 | 376 |

Table 3: Hyperparameters of AFedPG and the MLP policy parameterization settings.

| Hyperparameter | Setting | | | |
|---|---|---|---|---|
| Task | Swimmer-v4 | Hopper-v4 | Walker2D-v4 | Humanoid-v4 |
| MLP | $64 \times 64$ | $256 \times 256$ | $512 \times 512$ | $512 \times 512 \times 512$ |
| Activation function | ReLU | ReLU | ReLU | ReLU |
| Output function | Tanh | Tanh | Tanh | Tanh |
| Learning rate ($\alpha$) | $1 \times 10^{-3}$ | $1 \times 10^{-3}$ | $1 \times 10^{-3}$ | $1 \times 10^{-3}$ |
| Discount ($\gamma$) | 0.99 | 0.99 | 0.99 | 0.99 |
| Timesteps ($T$) | 2048 | 1024 | 1024 | 512 |
| Iterations ($K$) | $1 \times 10^{3}$ | $2 \times 10^{3}$ | $5 \times 10^{3}$ | $1.5 \times 10^{4}$ |
| Learning rate ($\eta$) | $3 \times 10^{-4}$ | $1 \times 10^{-4}$ | $5 \times 10^{-5}$ | $1 \times 10^{-5}$ |

A.3 SUPPLEMENTARY EXPERIMENTS

In this subsection, we show three experimental results to study the effect of computation heterogeneity, communication overhead, and reward performances with long runs.

**Effect of computation heterogeneity.** We study the effect of the computation heterogeneity among federated agents. The heterogeneity of computing powers is measured by the ratio $\frac{t_{\max}}{t_{\min}}$, and the effect is measured by the speedup, which is the global time of FedPG divided by that of AFedPG. Without loss of generality, we test the performances with (1) One straggler, which is one agent has $t_{\min}$ time consumption, and all the other agents have $t_{\max}$ time consumption. This is the scenario with the largest speedup. (2) One leader, which is one agent has $t_{\max}$ and the others have $t_{\min}$. This is the scenario with the smallest speedup.

In Figure 6, the results show that the speedup increases as the heterogeneity ratio $\frac{t_{\max}}{t_{\min}}$ increases. With one leader, as more agents participate, the speedup approximately decreases to a quadratic function w.r.t. the time heterogeneity ratio. With one straggler, the more agents that participate in the training process, the higher speedup they achieve. The overall results indicate that AFedPG has the potential to scale up to a very large RL system, particularly in scenarios with extreme stragglers (The ratio is large: $t_{\max} \gg t_{\min}$).

**Communication overhead analysis.** We compare the communication overhead of FedPG and AFedPG in Figure 7. For neural network parameters, we use the standard format float32. The communication overhead is measured by the number of transmitted bytes. The number of federated agents $N$ is set to 8. The MuJoCo task is Swimmer-v4. Overall, the commutative communication overhead is similar. However, when we dive deep into the agent side and the server side separately in fine-grained time, AFedPG shows advantages on both sides.

On the agent side, Figure 7 (a) shows the cumulative communication bytes during the training process. The cumulative communication overhead is similar in FedPG and AFedPG. In AFedPG, the faster ones, e.g., Agent 8, communicate more, and the slower ones, e.g., Agent 1, communicate

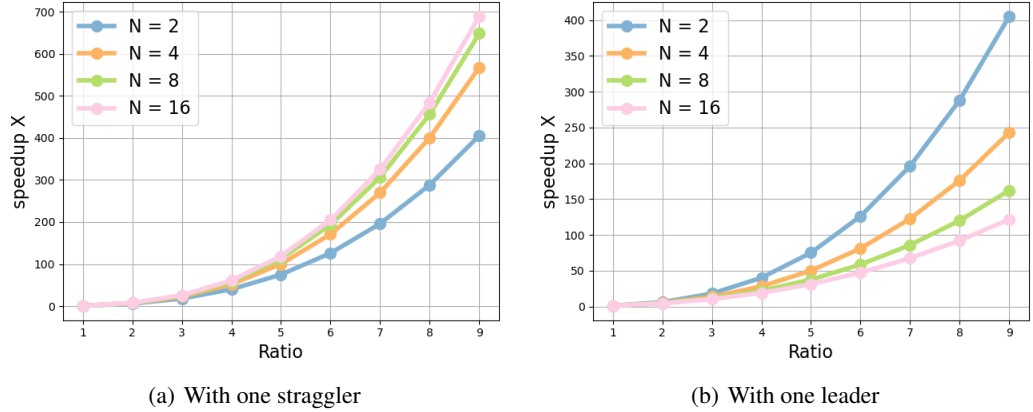

(a) With one straggler

(b) With one leader

Figure 6: The time complexity speedup of AFedPG compared to FedPG. $N$ is the number of agents. The x-axis is the heterogeneity ratio of computing power measured by $\frac{t_{\max}}{t_{\min}}$. The y-axis is the global time of FedPG divided by that of AFedPG.

less, which is more reasonable than the equal allocation in the synchronous setting. The Agent 1, in fact, commutes with the server during the training process, but it is insignificant in the plot with a log scale.

The agents have heterogeneous resources, but equal allocation could bring too much burden for the slower ones. In AFedPG, it naturally shifts these burdens to the faster ones considering the local resources.

On the server side, we show the downlink communication overhead in a time window in Figure 7 (b). The time window is set as "one global round in the synchronous setting". At time step 5, it is higher because two agents communicate with the server in a short time period. The total amounts of AFedPG and FedPG are similar. In AFedPG, it is almost evenly distributed during the time span, while in FedPG, the server has a huge burden with a peak. This makes the server in FedPG require huge resources.

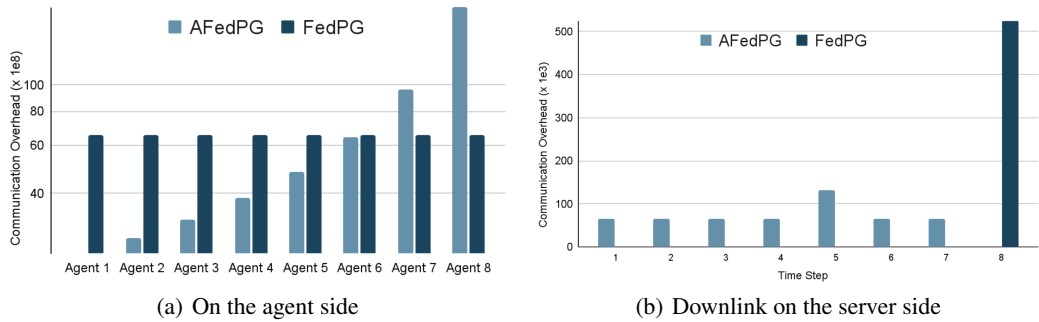

(a) On the agent side

(b) Downlink on the server side

Figure 7: The communication overhead of AFedPG compared to FedPG. The number of agents is $N = 8$. (a) The cumulative communication overhead on the agent side. (b) The downlink communication overhead in a time window on the server side.

**Reward performances with long runs.** To further enhance the results in Figure 3, we extend the running time with more samples. Compared to the results in Figure 3, we increase the number of samples by 5 times in each Mujoco task in Figure 8. The solid lines are averaged results over 5 runs with random seeds from 0 to 4. The shadowed areas are confidence intervals with 95% confidence level.

With enough samples, it basically achieves a similar reward performance for different numbers of agents $N$ in AFedPG. However, the more agents engage, the faster it achieves. Notably, the solid line is the average result with 5 independent runs. With different numbers of agents, the shadowed area has a large overlap. The more overlaps they have, the more runs that have similar reward performances because of the inherent randomness (with different random seeds) in the deep reinforcement learning tasks.

For the Humanoid-v4 task, though in some runs (shadowed area), PG achieves the optimal reward performance, the average (solid line) performance of PG is relatively lower than the others in AFedPG. The reason is that the Humanoid-v4 task has the largest state and action space, which makes the hyperparameter tuning difficult, $e.g.$, learning rates, and brings huge GPU hours. The hyperparameter setting of PG is suboptimal here, as it has no contribution to our main claim. Recall that we aim to use these experiments to verify the speedup effect in AFedPG in Table 1. The suboptimal hyperparameters of PG in the Humanoid-v4 task do not influence the conclusion: The more agents in AFedPG, the faster the optimal reward will be achieved.

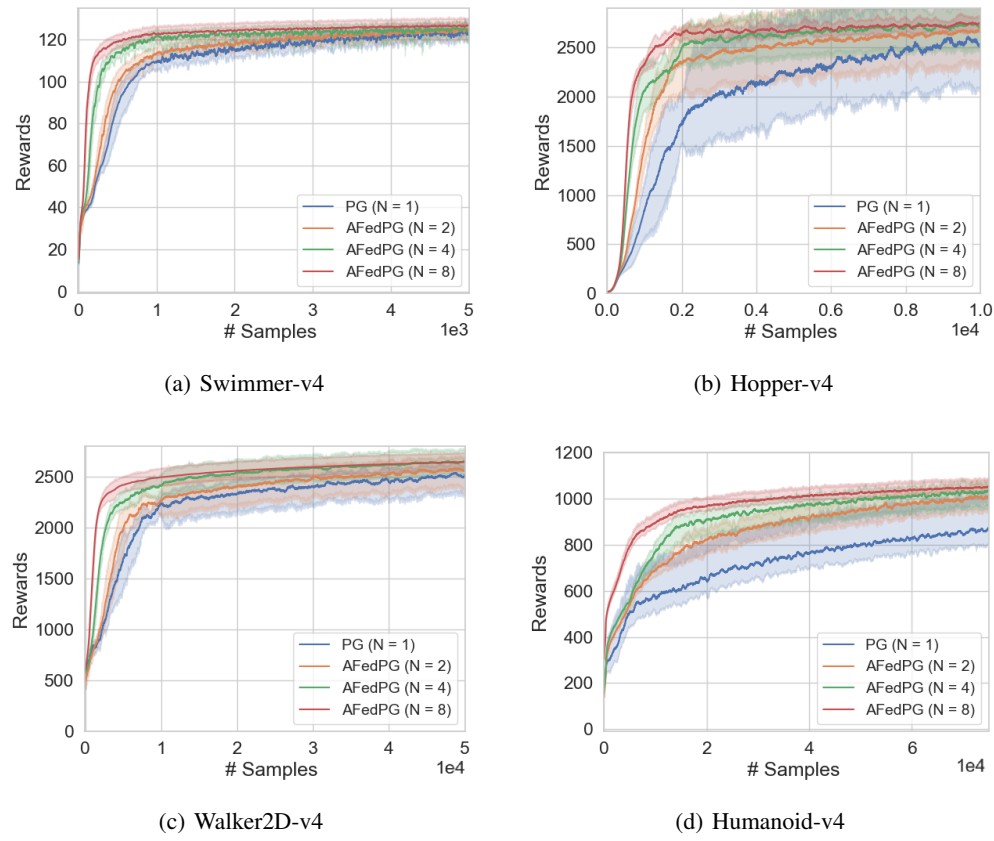

(a) Swimmer-v4

(b) Hopper-v4

(c) Walker2D-v4

(d) Humanoid-v4

Figure 8: Reward performances of AFedPG ($N = 2, 4, 8$) and PG ($N = 1$) on various MuJoCo environments, where $N$ is the number of federated agents. The solid lines are averaged results over 5 runs with random seeds from 0 to 4. The shadowed areas are confidence intervals with 95% confidence level.

## B  THEORETICAL PROOFS

In this section, we give the assumptions in Appendix B.1, the technical lemmas in Appendix B.2, our key lemmas in Appendix B.3, the proof of Theorem 5.1 (the global convergence) in Appendix B.4, and the proof of Theorem 5.2 (the FOSP convergence) in Appendix B.5.

### B.1  ASSUMPTIONS

In order to derive the global convergence rates, we make the following standard assumptions (Agarwal et al., 2021a; Ding et al., 2020; Liu et al., 2020; Papini et al., 2018; Xu et al., 2020) on policy gradients, and rewards.

**Assumption B.1.**

1. The score function is bounded as $\|\nabla \log \pi_\theta(a \mid s)\| \leq M_g$, for all $\theta \in \mathbb{R}^d$, $s \in \mathcal{S}$, and $a \in \mathcal{A}$.

2. The score function is $M_h$-Lipschitz continuous. In other words, for all $\theta_i$, $\theta_j \in \mathbb{R}^d$, $s \in \mathcal{S}$, and $a \in \mathcal{A}$, we have

$$\left\|\nabla \log \pi_{\theta_i}(a \mid s) - \nabla \log \pi_{\theta_j}(a \mid s)\right\| \leq M_h \left\|\theta_i - \theta_j\right\|. \tag{12}$$

3. The reward function is bounded as $r(s, a) \in [0, R]$, for all $s \in \mathcal{S}$, and $a \in \mathcal{A}$.

These standard assumptions naturally state that the reward, the first-order, and the second-order derivatives of the score function are not infinite, and they hold with common practical parametrization methods, *e.g.*, softmax policies. With function approximation $\pi_\theta$, the approximation error may not be 0 in practice.

Thus, we follow previous works Liu et al. (2020); Agarwal et al. (2021a); Ding et al. (2022); Huang et al. (2020) with an assumption on the expressivity of the policy parameterization class.

**Assumption B.2.** (Function approximation) $\exists \epsilon_{\text{bias}} \geq 0$ *s.t.* for all $\theta \in \mathbb{R}^d$, the transfer error satisfies

$$\mathbb{E}\left[\left(A_{\pi_\theta}(s, a) - (1 - \gamma)u^\star(\theta)^\top \nabla \log \pi_\theta(a \mid s)\right)^2\right] \leq \epsilon_{\text{bias}}, \tag{13}$$

where $u^\star(\theta) := F_\rho(\theta)^\dagger \nabla J(\theta)$, and $F_\rho(\theta)^\dagger$ is the Moore-Penrose pseudo-inverse of the Fisher matrix $F_\rho$.

It means that the parameterized policy $\pi_\theta$ makes the advantage function $A_{\pi_\theta}(s, a)$ approximated by the score function $\nabla \log \pi_\theta(a \mid s)$ as the features. This assumption is widely used with Fisher-non-degenerate parameterization. $\epsilon_{\text{bias}}$ can be very small with rich neural network parameterization, and 0 with a soft-max parameterization (Wang et al., 2020).

To achieve the global convergence, we make a standard assumption on the Fisher information matrix (Liu et al., 2020; Ding et al., 2022; Lan et al., 2023b).

**Assumption B.3.** (Positive definite) For all $\theta \in \mathbb{R}^d$, there exists a constant $\mu_F > 0$ *s.t.* the Fisher information matrix $F_\rho(\theta)$ induced by the policy $\pi_\theta$ and the initial state distribution $\rho$ satisfies

$$F_\rho(\theta) \succcurlyeq \mu_F \cdot I, \tag{14}$$

where $I \in \mathbb{R}^{d \times d}$ is an identity matrix.

For any two symmetric matrices $A$ and $B$ with the same dimension, $A \succcurlyeq B$ denotes that the eigenvalues of $A - B$ are greater or equal to zero.

### B.2  TECHNICAL LEMMAS

**Lemma B.4.** *For arbitrary $n$ vectors $\{\mathbf{a}_i \in \mathbb{R}^d\}_{i=1}^n$, we have*

$$\|\sum_{i=1}^n \mathbf{a}_i\|^2 \leq n \sum_{i=1}^n \|\mathbf{a}_i\|^2. \tag{15}$$

**Lemma B.5.** *Let $\alpha_k = (\frac{c}{t+c})^p$. $\forall p \in [0, 1]$ and $c \geq 1$, we have*

$$1 - \alpha_{k+1} \leq \frac{\alpha_{k+1}}{\alpha_k}. \tag{16}$$

*Proof.*

$$
\begin{aligned}
1 - \alpha_{k+1} &= 1 - (\frac{c}{t+1+c})^p \\
&\leq 1 - \frac{1}{t+1+c} \\
&\leq \frac{\alpha_{k+1}}{\alpha_k}.
\end{aligned} \tag{17}
$$

Straightforwardly, we have

$$\prod_{i=k}^{K-1} (1 - \alpha_{i+1}) \leq \frac{\alpha_K}{\alpha_k}. \tag{18}$$

$\square$

**Lemma B.6.** *Let $\alpha_k = (\frac{1}{k+1})^p$ and $\eta_k = \eta_0 (\frac{1}{k+1})^q$. $\forall p \in [0, 1)$, $q \geq 0$ and $\eta_0 \geq 0$, we have*

$$\sum_{k=0}^{K-1} \eta_k \prod_{i=k+1}^{K-1} (1 - \alpha_i) \leq c(p, q) \frac{\eta_K}{\alpha_K}, \tag{19}$$

*where $c(p, q) := \frac{2^{q-p}}{1-p} a \exp\left((1-p)2^p a^{1-p}\right)$ is a constant with specified $p$ and $q$, and $a = \max\left((\frac{q}{(1-p)2^p})^{\frac{1}{1-p}}, (\frac{2(q-p)}{(1-p)^2})^{\frac{1}{1-p}}\right)$.*

## B.3 KEY LEMMAS

In this subsection, we list four useful (key) lemmas to construct the proofs of Theorem 5.1 and Theorem 5.2.

Under Assumption B.1 on score functions, the following lemma holds based on the results (Lemma 5.4) in Zhang et al. (2020).

**Lemma B.7.** *The gradient of the expected return is $L_g$-continuous and $L_h$-smooth as follows*

$$
\begin{aligned}
\|\nabla J(\theta) - \nabla J(\theta')\| &\leq L_g \|\theta - \theta'\|, \\
\|\nabla^2 J(\theta) - \nabla^2 J(\theta')\| &\leq L_h \|\theta - \theta'\|,
\end{aligned} \tag{20}
$$

*where $L_g := \frac{R(M_g^2 + M_h)}{(1-\gamma)^2}$ and $L_h := \frac{RM_g^3(1+\gamma)}{(1-\gamma)^3} + \frac{RM_g M_h}{(1-\gamma)^2} + \mathcal{O}\left((1-\gamma)^{-1}\right)$.*

Under Assumption B.1, B.2 and B.3, we utilize the result (Lemma 4.7) in Ding et al. (2022) as the Lemma B.8.

**Lemma B.8.** *(Relaxed weak gradient domination) Under Assumptions B.1, B.2 and B.3, it holds that*

$$\|\nabla J(\theta)\| + \epsilon_g \geq \sqrt{2\mu(J^\star - J(\theta))}, \tag{21}$$

*where $\epsilon_g = \frac{\mu_F \sqrt{\epsilon_{\text{bias}}}}{M_g(1-\gamma)}$ and $\mu = \frac{\mu_F^2}{2M_g^2}$.*

Based on Lemma B.7, we derive our milestone (new), the ascent lemma with delayed updates, as follows:

**Lemma B.9.** *(Ascent Lemma with Delay) Under Assumptions B.1, it holds that*

$$-J(\theta_{k+1}) \leq -J(\theta_k) - \frac{1}{3}\eta_k \|\nabla J(\theta_k)\| + \frac{8}{3}\eta_k \|e_k\| + \frac{L_g}{2}\eta_k^2, \tag{22}$$

*where $e_k := d_{k-\delta_k} - \nabla J(\theta_k)$.*

This ascent lemma is specific to the asynchronous setting, which constructs the lower bound for the global increment, $J(\theta_{k+1}) - J(\theta_k)$, through the normalized policy gradients and our updating rules with delay $\delta_k$.

*Proof.* With the smoothness of the expected return $J(\theta)$ and the updating rule, we have

$$
\begin{aligned}
-J(\theta_{k+1}) &\leq -J(\theta_k) - \langle \nabla J(\theta_k), \theta_{k+1} - \theta_k \rangle + \frac{L_g}{2} \|\theta_{k+1} - \theta_k\|^2 \\
&= -J(\theta_k) - \eta_k \frac{\langle \nabla J(\theta_k), d_{k-\delta_k} \rangle}{\|d_{k-\delta_k}\|} + \frac{L_g}{2} \eta_k^2.
\end{aligned}
\tag{23}
$$

If $\|e_k\| \leq \frac{1}{2} \|\nabla J(\theta_k)\|$, we have

$$
\begin{aligned}
-\frac{\langle \nabla J(\theta_k), d_{k-\delta_k} \rangle}{\|d_{k-\delta_k}\|} &= -\frac{\|\nabla J(\theta_k)\|^2 + \langle \nabla J(\theta_k), e_k \rangle}{\|d_{k-\delta_k}\|} \\
&\leq \frac{-\|\nabla J(\theta_k)\|^2 + \|\nabla J(\theta_k)\|\|e_k\|}{\|d_{k-\delta_k}\|} \\
&\leq \frac{-\|\nabla J(\theta_k)\|^2 + \frac{1}{2}\|\nabla J(\theta_k)\|^2}{\|\nabla J(\theta_k)\| + \|e_k\|} \\
&\leq -\frac{1}{3} \|\nabla J(\theta_k)\|.
\end{aligned}
\tag{24}
$$

If $\|e_k\| \geq \frac{1}{2} \|\nabla J(\theta_k)\|$, we have

$$
\begin{aligned}
-\frac{\langle \nabla J(\theta_k), d_{k-\delta_k} \rangle}{\|d_{k-\delta_k}\|} &\leq \|\nabla J(\theta_k)\| \\
&= -\frac{1}{3} \|\nabla J(\theta_k)\| + \frac{4}{3} \|\nabla J(\theta_k)\| \\
&\leq -\frac{1}{3} \|\nabla J(\theta_k)\| + \frac{8}{3} \|e_k\|.
\end{aligned}
\tag{25}
$$

Combining these two conditions, and plugging the result into equation 23, the lemma can be proved as follows

$$
\begin{aligned}
-J(\theta_{k+1}) &\leq -J(\theta_k) - \eta_k \frac{\langle \nabla J(\theta_k), d_{k-\delta_k} \rangle}{\|d_{k-\delta_k}\|} + \frac{L_g}{2} \eta_k^2 \\
&\leq -J(\theta_k) - \frac{1}{3} \eta_k \|\nabla J(\theta_k)\| + \frac{8}{3} \eta_k \|e_k\| + \frac{L_g}{2} \eta_k^2.
\end{aligned}
\tag{26}
$$

$\square$

Next, we construct the relationship between the average concurrency $\bar{\omega}$ and the average delay $\bar{\delta}$ in Lemma B.10. This lemma gives the boundary (our result) of delays as a corollary of the result in Koloskova et al. (2022).

**Lemma B.10.** *The average delay $\bar{\delta}$ depends on the average concurrency $\bar{\omega}$, and they can be upper bounded as*

$$
\bar{\delta} = \frac{K+1}{K-1+|\mathcal{C}_K|} \bar{\omega} \leq \bar{\omega} \leq N.
\tag{27}
$$

*Proof.* Recall that $\{\delta_k^i\}_{i \in \mathcal{C}_k \setminus \{j_k\}}$ is the set of delays at the $k$-th global steps. After one global step, the number of cumulative delays over all agents increases by the current concurrency. Thus, we have the following connection

$$
\sum_{i=0}^{k} \delta_i + \sum_{i \in \mathcal{C}_{k+1} \setminus \{j_{k+1}\}} \delta_{k+1}^i = \sum_{i=0}^{k-1} \delta_i + \sum_{i \in \mathcal{C}_k \setminus \{j_k\}} \delta_k^i + \omega_{k+1}.
\tag{28}
$$

We note that there is no delay at the initial step (0-th iteration) of the algorithm. Therefore, we have $\delta_0^i = 0$ for all agents. Unrolling the above expression, at the $K$-th step, we have

$$\sum_{i=0}^{K-1} \delta_i + \sum_{i \in \mathcal{C}_K \setminus \{j_K\}} \delta_K^i = \sum_{i=0}^{K} \omega_{k+1} = (K+1)\bar{\omega}. \tag{29}$$

According to equation 7 and $|\mathcal{C}_K| \geq 2$, we achieve

$$\bar{\delta} = \frac{K+1}{K-1+|\mathcal{C}_K|}\bar{\omega} \leq \bar{\omega} \leq N. \tag{30}$$

$\square$

In practice, we need to use all the resources to speed up the training process. Thus, all agents engage in training, and $\bar{\omega} = \omega_{\max} = N$. Since the maximum concurrency is equal to the number of agents $N$, we have the upper boundary of the average delay as $\bar{\delta} \leq \omega_{\max} \leq N$.

Notably, we do not make any assumption on the largest delay $\delta_{\max}$, while we achieve the upper boundary of the average delay $\bar{\delta}$.

### B.4 Proof of Theorem 5.1 (Global Convergence Rate)

Under Assumption B.1 and Assumption B.2, we derive the global convergence rate of the proposed AFedPG.

First, we denote the difference between the policy gradient estimation $g(\widetilde{\tau}_k, \widetilde{\theta}_k)$ and the true policy gradient as

$$\xi_k := g(\widetilde{\tau}_k, \widetilde{\theta}_k) - \nabla J(\widetilde{\theta}_k), \tag{31}$$

and the expectation of the norm is bounded by $\sigma_g$.

Second, according to the updating rules in Algorithm 1 and Algorithm 2 , we expand the error term in Lemma B.9 as follows

$$
\begin{aligned}
e_k &= (1 - \alpha_{k-\delta_k})d_{k-1-\delta_{k-1}} - \nabla J(\theta_k) + \alpha_{k-\delta_k}g(\widetilde{\tau}_{k-\delta_k}, \widetilde{\theta}_{k-\delta_k}) \\
&= (1 - \alpha_{k-\delta_k})\big(d_{k-1-\delta_{k-1}} - \nabla J(\theta_{k-1})\big) + (1 - \alpha_{k-\delta_k})\big(\nabla J(\theta_{k-1}) - \nabla J(\theta_k)\big) \\
&\quad + \alpha_{k-\delta_k}\big(g(\widetilde{\tau}_{k-\delta_k}, \widetilde{\theta}_{k-\delta_k}) - \nabla J(\theta_k)\big) \\
&= \alpha_{k-\delta_k}\xi_{k-\delta_k} + (1 - \alpha_{k-\delta_k})e_{k-1} + (1 - \alpha_{k-\delta_k})\big(\nabla J(\theta_{k-1}) - \nabla J(\theta_k)\big) \\
&\quad + \alpha_{k-\delta_k}\big(\nabla J(\widetilde{\theta}_{k-\delta_k}) - \nabla J(\theta_k)\big) \\
&= \alpha_{k-\delta_k}\xi_{k-\delta_k} + (1 - \alpha_{k-\delta_k})e_{k-1} + (1 - \alpha_{k-\delta_k})\big(\nabla J(\theta_{k-1}) - \nabla J(\theta_k)\big) \\
&\quad + \alpha_{k-\delta_k}\big(\nabla J(\widetilde{\theta}_{k-\delta_k}) - \nabla J(\widetilde{\theta}_k)\big) + \alpha_{k-\delta_k}\big(\nabla J(\widetilde{\theta}_k) - \nabla J(\theta_k)\big) \\
&= \alpha_{k-\delta_k}\xi_{k-\delta_k} + (1 - \alpha_{k-\delta_k})e_{k-1} + \alpha_{k-\delta_k}\big(\nabla J(\widetilde{\theta}_{k-\delta_k}) - \nabla J(\widetilde{\theta}_k)\big) \\
&\quad + (1 - \alpha_{k-\delta_k})\big(\nabla J(\theta_{k-1}) - \nabla J(\theta_k) + \nabla^2 J(\theta_k)(\theta_{k-1} - \theta_k)\big) \\
&\quad + \alpha_{k-\delta_k}\big(\nabla J(\widetilde{\theta}_k) - \nabla J(\theta_k) + \nabla^2 J(\theta_k)(\theta_{k-1} - \theta_k)\big) \\
&\quad {\color{blue} -(1 - \alpha_{k-\delta_k})\nabla^2 J(\theta_k)(\theta_{k-1} - \theta_k) - \alpha_{k-\delta_k}\nabla^2 J(\theta_k)(\widetilde{\theta}_k - \theta_k)} \\
&\overset{8}{=} \alpha_{k-\delta_k}\xi_{k-\delta_k} + (1 - \alpha_{k-\delta_k})e_{k-1} + \alpha_{k-\delta_k}\big(\nabla J(\widetilde{\theta}_{k-\delta_k}) - \nabla J(\widetilde{\theta}_k)\big) \\
&\quad + (1 - \alpha_{k-\delta_k})\big(\nabla J(\theta_{k-1}) - \nabla J(\theta_k) + \nabla^2 J(\theta_k)(\theta_{k-1} - \theta_k)\big) \\
&\quad + \alpha_{k-\delta_k}\big(\nabla J(\widetilde{\theta}_k) - \nabla J(\theta_k) + \nabla^2 J(\theta_k)(\theta_{k-1} - \theta_k)\big).
\end{aligned}
\tag{32}
$$

Thus, the error term $e_k$ can be written in a recursive way (contains $e_{k-1}$) with serval terms that contain policy gradients. We aim to derive the upper boundary for each term at the next step.

**Remark**: The Hessian correction terms (in blue) are equal to 0 according to our updating rules (delay-adaptive lookahead update) in Step 8 of Algorithm 1. We design this technique to cancel out the second-order terms, and thus achieve the desired convergence rate. Without our technique, it would be hard to bound the above errors.

Next, we denote each term in equation 32 separately as follows

$$
\begin{aligned}
A_k &:= \nabla J(\theta_{k-1}) - \nabla J(\theta_k) + \nabla^2 J(\theta_k)(\theta_{k-1} - \theta_k), \\
B_k &:= \nabla J(\widetilde{\theta}_k) - \nabla J(\theta_k) + \nabla^2 J(\theta_k)(\theta_{k-1} - \theta_k), \\
C_k &:= \nabla J(\widetilde{\theta}_{k-\delta_k}) - \nabla J(\widetilde{\theta}_k).
\end{aligned}
\tag{33}
$$

Now, we start to bound each term. With the smoothness of the expected discounted return function in Lemma B.7 and the non-increasing learning rates, we have

$$
\begin{aligned}
\|A_k\| &\leq L_h \|\theta_{k-1} - \theta_k\|^2 = L_h \eta_{k-1}^2, \\
\|B_k\| &\leq L_h \|\widetilde{\theta}_k - \theta_k\|^2 = L_h \frac{(1 - \alpha_{k-\delta_k})^2}{\alpha_{k-\delta_k}^2} \eta_{k-1}^2, \\
\|C_k\| &\leq L_g \|\widetilde{\theta}_{k-\delta_k} - \widetilde{\theta}_k\| \\
&= L_g \left\| \sum_{i=k-\delta_k}^{k-1} \widetilde{\theta}_{i+1} - \widetilde{\theta}_i \right\| \\
&\leq L_g \sum_{i=k-\delta_k}^{k-1} \|\widetilde{\theta}_{i+1} - \widetilde{\theta}_i\| \\
&= L_g \sum_{i=k-\delta_k}^{k-1} \left\| \frac{1}{\alpha_{i+1-\delta_{i+1}}}(\theta_{i+1} - \theta_i) + \frac{1 - \alpha_{i-\delta_i}}{\alpha_{i-\delta_i}}(\theta_i - \theta_{i-1}) \right\| \\
&\leq L_g \sum_{i=k-\delta_k}^{k-1} \left\| \frac{1}{\alpha_{i+1-\delta_{i+1}}}(\theta_{i+1} - \theta_i) \right\| + \left\| \frac{1 - \alpha_{i-\delta_i}}{\alpha_{i-\delta_i}}(\theta_i - \theta_{i-1}) \right\| \\
&= L_g \sum_{i=k-\delta_k}^{k-1} \left( \frac{1}{\alpha_{i+1-\delta_{i+1}}} \eta_i + \frac{1 - \alpha_{i-\delta_i}}{\alpha_{i-\delta_i}} \eta_{i-1} \right) \\
&\leq L_g \sum_{i=k-\delta_k}^{k-1} \left( \frac{2}{\alpha_{k-1}} \eta_{k-\delta_k} \right) \\
&= \delta_k L_g \frac{2\eta_{k-\delta_k}}{\alpha_{k-1}},
\end{aligned}
\tag{34}
$$

where the equalities are simple plugins according to the updating rules in Algorithm 1. The last step in equation 34 happens, because there is no index $i$ inside the summation operation, and it sums a constant for $\delta_k$ times.

We denote $\beta_k := \prod_{i=k+1}^{K}(1 - \alpha_{i-\delta_i})$ with $\beta_K = 1$. Next, unrolling the recursion equation 32, we have the error at the $K$-th step as follows

$$
e_K = \beta_0 e_0 + \sum_{k=1}^{K} \alpha_{k-\delta_k} \beta_k \xi_{k-\delta_k} + \sum_{k=1}^{K}(1 - \alpha_{k-\delta_k})\beta_k A_k + \sum_{k=1}^{K} \alpha_{k-\delta_k} \beta_k (B_k + C_k).
\tag{35}
$$

Choose learning rates $\alpha_k = (\frac{1}{k+1})^{\frac{4}{5}}$ and $\eta_k = \eta_0 \frac{1}{k+1}$, where $\eta_0$ is a constant and we will show the value later. Using Jensen's inequality and technical lemmas, we achieve the following bound

$$
\begin{aligned}
\mathbb{E}[\|e_K\|] &\leq \beta_0 \mathbb{E}[\|e_0\|] + \left( \mathbb{E}\left[ \left\| \sum_{k=1}^{K} \alpha_{k-\delta_k} \beta_k \xi_{k-\delta_k} \right\|^2 \right] \right)^{\frac{1}{2}} + \sum_{k=1}^{K} (1 - \alpha_{k-\delta_k}) \beta_k \mathbb{E}[\|A_k\|] \\
&\quad + \sum_{k=1}^{K} \alpha_{k-\delta_k} \beta_k (\mathbb{E}[\|B_k\|] + \mathbb{E}[\|C_k\|]) \\
&\leq \beta_0 \mathbb{E}[\|e_0\|] + \left( \sum_{k=1}^{K} \alpha_{k-\delta_k}^2 \beta_k^2 \mathbb{E}\left[ \|\xi_{k-\delta_k}\|^2 \right] \right)^{\frac{1}{2}} + \sum_{k=1}^{K} (1 - \alpha_{k-\delta_k}) \beta_k \mathbb{E}[\|A_k\|] \\
&\quad + \sum_{k=1}^{K} \alpha_{k-\delta_k} \beta_k (\mathbb{E}[\|B_k\|] + \mathbb{E}[\|C_k\|]) \\
&\overset{34}{\leq} \beta_0 \sigma_g + \left( \sum_{k=1}^{K} \alpha_{k-\delta_k}^2 \beta_k^2 \mathbb{E}\left[ \|\xi_{k-\delta_k}\|^2 \right] \right)^{\frac{1}{2}} + L_h \sum_{k=1}^{K} (1 - \alpha_{k-\delta_k}) \beta_k \eta_{k-1}^2 \\
&\quad + L_h \sum_{k=1}^{K} \beta_k (1 - \alpha_{k-\delta_k})^2 \frac{\eta_{k-1}^2}{\alpha_{k-\delta_k}} + 2 L_g \sum_{k=1}^{K} \alpha_{k-\delta_k} \beta_k \delta_k \frac{\eta_{k-\delta_k}^2}{\alpha_{k-1}} \\
&\leq \beta_0 \sigma_g + \left( \sum_{k=1}^{K} \alpha_{k-\delta_k}^2 \beta_k^2 \mathbb{E}\left[ \|\xi_{k-\delta_k}\|^2 \right] \right)^{\frac{1}{2}} + L_h \sum_{k=1}^{K} \beta_k \frac{\eta_{k-1}^2}{\alpha_{k-\delta_k}} \\
&\quad + 2 L_g \sum_{k=1}^{K} \alpha_{k-\delta_k} \beta_k \delta_k \frac{\eta_{k-\delta_k}^2}{\alpha_{k-1}} \\
&\leq \beta_0 \sigma_g + \left( \sum_{k=1}^{K} \alpha_{k-\delta_k}^2 \beta_k^2 \right)^{\frac{1}{2}} \sigma_g + L_h \sum_{k=1}^{K} \beta_k \frac{\eta_{k-1}^2}{\alpha_{k-\delta_k}} + 2 L_g \sum_{k=1}^{K} \alpha_{k-\delta_k} \beta_k \delta_k \frac{\eta_{k-\delta_k}^2}{\alpha_{k-1}} \\
&\leq \beta_0 \sigma_g + \left( \sum_{k=1}^{K} \alpha_{k-\delta_k}^2 \beta_k^2 \right)^{\frac{1}{2}} \sigma_g + \frac{9 L_h}{4} \sum_{k=1}^{K} \beta_k \frac{\eta_k^2}{\alpha_k} + 2 L_g \sum_{k=1}^{K} \alpha_{k-\delta_k} \beta_k \delta_k \frac{\eta_{k-\delta_k}^2}{\alpha_{k-1}} \\
&\overset{37,39,40,41}{\leq} \alpha_K \sigma_g + c_1 \sqrt{\alpha_K} \sigma_g + \frac{9}{4} c_2 L_h \frac{\eta_K^2}{\alpha_K^2} + c_3 L_g \frac{\eta_K^2}{\alpha_K^{1.75}} \bar{\delta},
\end{aligned}
\tag{36}
$$

where $c_1 := 2\sqrt{c(\frac{4}{5}, \frac{4}{5})}$, $c_2 := c(\frac{4}{5}, \frac{6}{5})$, $c_3 := 8\sqrt{c(\frac{4}{5}, \frac{16}{5})}$ are constants, and the values of $c(\cdot, \cdot)$ are defined in Lemma B.6.

Notably, as we state in the last paragraph in Section 5, $t_i$ is pre-determined by the computation resource at agent $i$ and is fixed. Thus, the delay $\delta_k$ is pre-determined by the system. It is not a random variable, but unknown until given the exact system setting. This allows us to derive the first inequality in equation 36. This pre-determined property is also suitable for all the derivations below.

We explain the boundary derivation details here. We first derive the first term of the boundary in equation 36 as follows

$$
\begin{aligned}
\beta_0 &= \prod_{i=1}^{K}(1 - \alpha_{i-\delta_i}) \\
&\leq \prod_{i=1}^{K}(1 - \alpha_i) \\
&= \prod_{i=0}^{K-1}(1 - \alpha_{i+1}) \\
&\overset{18}{\leq} \frac{\alpha_K}{\alpha_0} \\
&= \alpha_K.
\end{aligned}
\tag{37}
$$

In the training process, at step $k$, when an agent sends the update to the server, as long as the agent communicates with the server during the last half training process, the delay $\delta_k \leq \frac{k}{2}$. This becomes almost certain when $k$ is large and $k$ is usually much larger than the upper bound of the currency $N$. Thus, we have

$$
\begin{aligned}
\alpha_{k-\delta_k} &= \left(\frac{1}{k+1-\delta_k}\right)^{\frac{4}{5}} \\
&= \left(\frac{1}{k+1}\right)^{\frac{4}{5}}\left(\frac{k+1}{k+1-\delta_k}\right)^{\frac{4}{5}} \\
&\leq \left(\frac{1}{k+1}\right)^{\frac{4}{5}}\left(\frac{k+1}{k+1-\frac{k}{2}}\right)^{\frac{4}{5}} \\
&\leq 2\left(\frac{1}{k+1}\right)^{\frac{4}{5}} \\
&= 2\alpha_k.
\end{aligned}
\tag{38}
$$

We then derive the second term of the boundary in equation 36 as follows

$$
\begin{aligned}
\sum_{k=1}^{K}\alpha_{k-\delta_k}^2\beta_k^2 &\overset{38}{\leq} 4\sum_{k=1}^{K}\alpha_k^2\beta_k^2 \\
&= 4\sum_{k=1}^{K}\alpha_k^2\prod_{i=k+1}^{K}(1-\alpha_{i-\delta_i})\prod_{i=k+1}^{K}(1-\alpha_{i-\delta_i}) \\
&\leq 4\sum_{k=1}^{K}\alpha_k^2\prod_{i=k+1}^{K}(1-\alpha_i)\prod_{i=k+1}^{K}(1-\alpha_i) \\
&\leq 4\sum_{k=1}^{K}\alpha_k^2\prod_{i=k+1}^{K-1}(1-\alpha_i)\prod_{i=k}^{K-1}(1-\alpha_{i+1}) \\
&\overset{18}{\leq} 4\sum_{k=1}^{K}\alpha_k^2\prod_{i=k+1}^{K-1}(1-\alpha_i)\frac{\alpha_K}{\alpha_k} \\
&= 4\alpha_K\sum_{k=1}^{K}\alpha_k\prod_{i=k+1}^{K-1}(1-\alpha_i) \\
&\leq 4\alpha_K\sum_{k=0}^{K-1}\alpha_k\prod_{i=k+1}^{K-1}(1-\alpha_i) \\
&\overset{19}{\leq} 4\alpha_Kc(\frac{4}{5},\frac{4}{5})\frac{\alpha_K}{\alpha_K} \\
&= 4\alpha_Kc(\frac{4}{5},\frac{4}{5}).
\end{aligned}
\tag{39}
$$

Next, we derive the third term of the boundary in equation 36 as follows

$$
\begin{aligned}
\sum_{k=1}^{K} \frac{\eta_k^2}{\alpha_k} \beta_k &= \eta_0^2 \sum_{k=1}^{K} \left(\frac{1}{k+1}\right)^{\frac{6}{5}} \beta_k \\
&= \eta_0^2 \sum_{k=1}^{K} \left(\frac{1}{k+1}\right)^{\frac{6}{5}} \prod_{i=k+1}^{K} (1 - \alpha_{i-\delta_i}) \\
&\leq \eta_0^2 \sum_{k=1}^{K} \left(\frac{1}{k+1}\right)^{\frac{6}{5}} \prod_{i=k+1}^{K} (1 - \alpha_i) \\
&\leq \eta_0^2 \sum_{k=1}^{K} \left(\frac{1}{k+1}\right)^{\frac{6}{5}} \prod_{i=k+1}^{K-1} (1 - \alpha_i) \\
&\leq \eta_0^2 \sum_{k=0}^{K-1} \left(\frac{1}{k+1}\right)^{\frac{6}{5}} \prod_{i=k+1}^{K-1} (1 - \alpha_i) \\
&\overset{19}{\leq} \eta_0^2 c\left(\frac{4}{5}, \frac{6}{5}\right) \left(\frac{1}{K+1}\right)^{\frac{2}{5}} \\
&= c\left(\frac{4}{5}, \frac{6}{5}\right) \frac{\eta_K^2}{\alpha_K^2}.
\end{aligned}
\tag{40}
$$

At last, we derive the fourth term of the boundary in equation 36. We use Cauchy–Schwarz inequality and the fact that the second norm of a vector is always equal or smaller than the first norm.

$$
\begin{aligned}
\sum_{k=1}^{K} \alpha_{k-\delta_k} \beta_k \delta_k \frac{\eta_{k-\delta_k}^2}{\alpha_{k-1}} &\leq \sum_{k=1}^{K} \eta_{k-\delta_k}^2 \beta_k \delta_k \\
&= \sum_{k=1}^{K} \eta_k^2 \left(\frac{k+1}{k+1-\frac{k}{2}}\right)^2 \beta_k \delta_k \\
&\leq 4\eta_0^2 \sum_{k=1}^{K} \left(\frac{1}{k+1}\right)^2 \beta_k \delta_k \\
&\leq 4\eta_0^2 \left(\sum_{k=1}^{K} \left(\frac{1}{k+1}\right)^4 \beta_k^2 \cdot \sum_{k=1}^{K} \delta_k^2\right)^{\frac{1}{2}} \\
&\leq 4\eta_0^2 \left(\sum_{k=1}^{K} \left(\frac{1}{k+1}\right)^4 \beta_k^2\right)^{\frac{1}{2}} \sum_{k=1}^{K} \delta_k \\
&\leq 4\eta_0^2 K\bar{\delta} \left(\sum_{k=1}^{K} \left(\frac{1}{k+1}\right)^4 \beta_k^2\right)^{\frac{1}{2}} \\
&\leq 4\eta_0^2 K\bar{\delta} \left(\sum_{k=1}^{K} \left(\frac{1}{k+1}\right)^4 \beta_k \prod_{i=k+1}^{K} (1 - \alpha_{i-\delta_i})\right)^{\frac{1}{2}}
\end{aligned}
$$

$$
\begin{aligned}
&\leq 4\eta_0^2 K\bar{\delta}\left(\sum_{k=1}^{K}\left(\frac{1}{k+1}\right)^4\beta_k\frac{\alpha_K}{\alpha_k}\right)^{\frac{1}{2}}\\
&= 4\eta_0^2 K\bar{\delta}\left(\alpha_K\sum_{k=1}^{K}\left(\frac{1}{k+1}\right)^{\frac{16}{5}}\beta_k\right)^{\frac{1}{2}}\\
&= 4\eta_0^2 K\bar{\delta}\left(\alpha_K\sum_{k=1}^{K}\left(\frac{1}{k+1}\right)^{\frac{16}{5}}\prod_{i=k+1}^{K}(1-\alpha_{i-\delta_i})\right)^{\frac{1}{2}}\\
&\leq 4\eta_0^2 K\bar{\delta}\left(\alpha_K\sum_{k=1}^{K}\left(\frac{1}{k+1}\right)^{\frac{16}{5}}\prod_{i=k+1}^{K}(1-\alpha_i)\right)^{\frac{1}{2}}\\
&\leq 4\eta_0^2 K\bar{\delta}\left(\alpha_K\sum_{k=1}^{K}\left(\frac{1}{k+1}\right)^{\frac{16}{5}}\prod_{i=k+1}^{K-1}(1-\alpha_i)\right)^{\frac{1}{2}}\\
&\overset{19}{\leq} 4\eta_0^2 K\bar{\delta}\left(\alpha_K c(\tfrac{4}{5},\tfrac{16}{5})(\tfrac{1}{K+1})^{\frac{12}{5}}\right)^{\frac{1}{2}}\\
&\leq 4\eta_0^2 K\bar{\delta}\sqrt{c(\tfrac{4}{5},\tfrac{16}{5})}(\tfrac{1}{K+1})^{\frac{8}{5}}\\
&\leq 4\eta_0^2\bar{\delta}\sqrt{c(\tfrac{4}{5},\tfrac{16}{5})}(\tfrac{1}{K+1})^{\frac{3}{5}}\\
&= 4\bar{\delta}\sqrt{c(\tfrac{4}{5},\tfrac{16}{5})}\frac{\eta_K^2}{\alpha_K^{1.75}}.
\end{aligned}
\tag{41}
$$

Now, we derive the final convergence rate. After plugging the result into Lemma B.9 and using Lemma B.8 (Ascent Lemma with Delay), we achieve the following inequality

$$
\begin{aligned}
J^\star - \mathbb{E}[J(\theta_{k+1})] &\leq (1-\frac{\sqrt{2\mu}\eta_k}{3})\left(J^\star - \mathbb{E}[J(\theta_k)]\right) + \frac{\eta_k}{3}\epsilon_g + \frac{8}{3}\eta_k\mathbb{E}[\|e_k\|] + \frac{L_g}{2}\eta_k^2\\
&\overset{36}{\leq} (1-\frac{\sqrt{2\mu}\eta_k}{3})\left(J^\star - \mathbb{E}[J(\theta_k)]\right) + \frac{\eta_k}{3}\epsilon_g + \frac{L_g}{2}\eta_k^2\\
&\quad + \frac{8}{3}\eta_k\left(\alpha_k\sigma_g + c_1\sqrt{\alpha_k}\sigma_g + \frac{9}{4}c_2 L_h\frac{\eta_k^2}{\alpha_k^2} + c_3 L_g\frac{\eta_k^2}{\alpha_k^{1.75}}\bar{\delta}\right).
\end{aligned}
\tag{42}
$$

Unrolling this recursion, we have

$$
\begin{aligned}
J^\star - \mathbb{E}[J(\theta_K)] &\leq \frac{\eta_0}{3}\epsilon_g + \frac{J^\star - \mathbb{E}[J(\theta_0)]}{(K+1)^2} + c_3\frac{8\eta_0^3}{3}\frac{L_g}{(K+1)^{\frac{3}{5}}}\bar{\delta} + \frac{\eta_0^2}{2}\frac{L_g}{K+1}\\
&\quad + \frac{8\eta_0}{3}\frac{\sigma_g}{(K+1)^{\frac{4}{5}}} + c_1\frac{8\eta_0}{3}\frac{\sigma_g}{(K+1)^{\frac{2}{5}}} + 6c_2\eta_0^3\frac{L_h}{(K+1)^{\frac{2}{5}}}.
\end{aligned}
\tag{43}
$$

Note that, introduced in Lemma B.7, the discount factor $\gamma$ is contained in $L_g = \mathcal{O}\left((1-\gamma)^{-2}\right)$ and $L_h = \mathcal{O}\left((1-\gamma)^{-3}\right)$. Comparing with the definition of $\epsilon_g$ in Lemma B.8, we choose $\eta_0 = \frac{3\mu_F}{M_g}$ for the criteria. Thus, to satisfy the global convergence criterion $J^\star - \mathbb{E}[J(\theta_K)] \leq \epsilon + \frac{\sqrt{\epsilon_{\text{bias}}}}{1-\gamma}$, we have the iteration complexity $K = \mathcal{O}(\epsilon^{-2.5})$. In our algorithms, the sample complexity is equal to the iteration complexity, which is also $\mathcal{O}(\epsilon^{-2.5})$.

## B.5 PROOF OF THEOREM 5.2 (FOSP CONVERGENCE RATE)

Under Assumption B.1 and Assumption B.3, we derive the first-order stationary convergence rate of AFedPG. Notably, the FOSP convergence does not require Assumption B.2, which makes assumptions on the neural network approximation error.

First, we denote the average of gradient expectations as

$$\mathbb{E}\|\nabla J(\bar{\theta}_K)\| := \frac{\sum_{k=1}^{K} \eta_k \mathbb{E}\|\nabla J(\theta_k)\|}{\sum_{k=1}^{K} \eta_k}. \tag{44}$$

Next, rearranging the terms in Lemma B.9 (Ascent Lemma with Delay) and summing up the inequality, we achieve the inequality below

$$\mathbb{E}\|\nabla J(\bar{\theta}_K)\| \leq \frac{3}{\sum_{k=1}^{K} \eta_k} \left( J^\star - \mathbb{E}[J(\theta_0)] \right) + \frac{8}{\sum_{k=1}^{K} \eta_k} \sum_{k=1}^{K} \eta_k \mathbb{E}[\|e_k\|] \\ + \frac{3L_g}{2\sum_{k=1}^{K} \eta_k} \sum_{k=1}^{K} \eta_k^2. \tag{45}$$

Choose learning rates $\alpha_k = (\frac{1}{k+1})^{\frac{4}{7}}$ and $\eta_k = \eta_0 (\frac{1}{k+1})^{\frac{5}{7}}$. Plug in the result into equation 36 that we have shown in Appendix B.4, we have

$$\mathbb{E}\|\nabla J(\bar{\theta}_K)\| \leq \frac{3(J^\star - \mathbb{E}[J(\theta_0)])}{(K+1)^{\frac{2}{7}}} + 16c_3\eta_0^3 \frac{L_g}{(K+1)^{\frac{3}{7}}}\bar{\delta} + \frac{3\eta_0 L_g}{(K+1)^{\frac{5}{7}}} \\ + 16\eta_0 \frac{\sigma_g}{(K+1)^{\frac{4}{7}}} + 16c_1\eta_0 \frac{\sigma_g}{(K+1)^{\frac{2}{7}}} + 36c_2\eta_0^3 \frac{L_h}{(K+1)^{\frac{2}{7}}}. \tag{46}$$

Thus, to satisfy the FOSP convergence criterion $\mathbb{E}\|\nabla J(\bar{\theta}_K)\| \leq \epsilon$, we have the iteration complexity $K = \mathcal{O}(\epsilon^{-3.5})$. In our algorithms, the sample complexity is equal to the iteration complexity, which is also $\mathcal{O}(\epsilon^{-3.5})$.

Notably, this result does not rely on Lemma B.8 and Assumption B.1. The norm of the average gradient is approaching an arbitrarily small value during the training process, regardless of the function approximation error $\epsilon_{\text{bias}}$.

# C  FURTHER DISCUSSIONS

## C.1  COMPARISON WITH PREVIOUS WORKS

**Comparison with** (Shen et al., 2023). We first acknowledge the theoretical contribution of this pioneer work. However, there are several limitations and differences.

1. (Different RL Algorithm Class) Instead of PG, Shen et al. (2023) is an actor-critic (AC) method with extra value networks, which requires much more computation and memory cost compared to the pure policy gradient (PG) method. Thus, the fine-tuning of Gemini (Team & DeepMind, 2024) and GPT-4 (OpenAI, 2023) uses PG methods instead of AC methods.

2. (General Function Parameterization) Shen et al. (2023) only has Linear Parameterization (Deep RL is not included.) for the global convergence analysis, which has limited practical meaning. With a General Function Parametrization, *e.g.*, neural networks (Deep RL), there is no such a result. We consider a general and practical setting with a General Function Parameterization in our work.

3. (Convergence Performance) Even in the single-agent setting (without federated agents), the SOTA result of the AC method is $\mathcal{O}(\epsilon^{-3})$ (Gaur et al., 2024) and the previous approach is $\mathcal{O}(\epsilon^{-6})$ (Fu et al., 2021), which is still worse than our $\mathcal{O}(\epsilon^{-2.5})$. In the federated setting, there is no result that achieves $\mathcal{O}(\epsilon^{-3})$ for AC methods. Moreover, with a general function parameterization, we compare the performances of their A3C in Figure 4, which is much worse.

4. (Assumptions) Shen et al. (2023) relies on a strong and unpractical assumption, their Assumption 2. It assumes that the largest delay is bounded by a constant $K_0$. However, in practice, the slowest agent may not communicate with the server, and thus, has an infinite delay. In our analysis, we do not require any boundary for the largest delay, because we only contain the average delay in the convergence rate, and the average delay is naturally bounded by the number of agents in Lemma B.10 (our corollary).

## C.2  DIFFERENCE BETWEEN FL, FEDRL, AND AFEDRL

Unlike supervised FL where local datasets are fixed or pre-specified, in RL, agents collect new samples in each iteration based on the current local policies. The new data are collected with dynamic dependencies, which do not appear in all prior FL and A-FL works.

In synchronous FedRL, each agent collects samples according to the same global policy $\pi_\theta$. However, in AFedRL, even if all agents have an identical environment, each agent collects samples according to different policies $\tau_k \sim p(\cdot|\pi_{\theta_k})$, because of the delay. This dynamic nature makes both the problem itself and the theoretical analysis challenging. We propose a new delay-adaptive model aggregation strategy to tackle these unique challenges of FedRL.

Moreover, data collected by each agent are naturally (non-manually controlled) heterogeneous. Despite having identical environments, agents collect data according to their own (different) policies, a fundamental difficulty that our AFedPG paper solves, as verified theoretically and empirically.

