# OpenReview forum: "Asynchronous Federated Reinforcement Learning with Policy Gradient Updates: Algorithm Design and Convergence Analysis"
_ICLR.cc/2025/Conference — ICLR 2025 Poster_

### Official Review · Reviewer_oswF · 2024-10-23

**Soundness:** 3
**Presentation:** 3
**Contribution:** 3
**Rating:** 8
**Confidence:** 4

**Summary:**

This paper proposes an asynchronous federated reinforcement learning framework termed AFedPG for the policy gradient algorithm. It designs a delay-adaptive lookahead technique that can effectively handle heterogeneous arrival times of policy gradients. This work shows theoretical linear speedup in terms of the norm for policy gradient and verifies the speedup effect numerically.

**Strengths:**

1. The proposed framework handles the delayed arrival of policy-gradient and reduces the waiting time compared to the algorithm for the homogeneous setting.

2. The authors propose their special step size designs to cancel out a second-order error term when conducting the error analysis, which serves as a technical novelty.

3. Numerical experiments demonstrate that the authors accelerate the training process compared to the synchronous algorithm.

**Weaknesses:**

1. Issues in Section 4. The authors are encouraged to explain more about the concepts of active agents, concurrency, and delay. In algorithm 2, the authors are encouraged to explain more details about model sharing from the central server as how the agents hold $d_{k-1}$ and $\theta_{k-1}$ is not explicitly explained. In addition, the authors are encouraged to explain the relationship between their algorithms and the single-agent and homogeneous counterparts in the literature. Last, the authors assume that the agents can sample a trajectory with infinite lengths, which is impossible in practice. The authors are recommended to explain more on such assumptions.

2. Issues in Section 5. (a) In equations 10 and 11, RHS contains a constant term that does not depend on $K$, which originates from the function approximation error as indicated in the appendix. The authors are encouraged to explain this term in the main paper. (b) The authors are encouraged to explain how they get the total waiting time in line 394.

3. Issues in Appendix B (proofs). (a) The authors are encouraged to explain more about the definitions and notations that are already established in the literature, for example, $F_\rho(\theta),\mu_F,\sigma_g$. (b) In Lemmas B.6 and B.7, the authors are recommended to point out the cited lemma in the references. (c) The second term in line 1084 should be $(\mathbb{E}\cdot^2)^{1/2}$. (d) In equations 37 and 38, there are typos related to $\nabla$. (e) In line 1028, there is a typo related to $d_{\delta_{k-1}}$.

**Questions:**

See the weakness.

---

> ### Author Response · Authors · 2024-11-24
>
> Thank you for reading the paper carefully! If there are any other concerns, we would be happy to answer them.
>
> ---
>
> **W1.** Section 4.
>
> We have added more explanation of $d_{k-1}$ and $\theta_{k-1}$, and reconstructed our algorithms (Algorithm 1 and 2) to clearly show the process. We keep all notations unchanged in the theoretical analysis.
>
> We have added literature explanations in the Introduction.
>
> For trajectory sampling, it is indeed not practical to sample the whole trajectory. However, with the discount factor $\gamma$, the variance is exponentially small in the horizon [1,2]. The truncation does not influence the convergence results, and thus, the infinite horizon setting [3] is widely used in the recent analysis.
>
> ---
>
> **W2.** Section 5.
>
> Thank you for the remainder! Though the equations are correct in the theorems in the appendix, we had a typo in its simplified version in equation 10. We have fixed the typo in the main paper.
>
> We have added the derivation with details in Appendix A.1. The line 394 could be well explained according to Appendix A.1.
>
> ---
>
> **W3.** Appendix B.
>
> Thanks for the suggestion! We have added the explanations of the conventional notations in Appendix B.1.
>
> We have added the exact positions of the cited lemmas (Lemma B.7 and B.8).
>
> We have fixed all typos with more rounds of proofreading.
>
> ---
>
> [1] On the Theory of Policy Gradient Methods: Optimality, Approximation, and Distribution Shift. Journal of Machine Learning Research 2021.
>
> [2] An Improved Analysis of (Variance-Reduced) Policy Gradient and Natural Policy Gradient Methods. NeurIPS 2020.
>
> [3] Improved Sample Complexity Analysis of Natural Policy Gradient Algorithm with General Parameterization for Infinite Horizon Discounted Reward Markov Decision Processes. AISTATS 2024.

---

> > ### Comment · Reviewer_oswF · 2024-11-26
> > **Reply to the authors**
> >
> > Thanks to the authors for their detailed replies and revision. My mentioned issues are fixed, and the algorithm is now in a clear state. Overall, I believe the heterogeneous updating scheme is very interesting. Such technical analysis is new to the existing literature and is of broader interest to the ICLR audience.
> >
> >
> > Thus, I increase my rating for Presentation to 3 and keep the overall rating of 6.

---

> > > ### Author Response · Authors · 2024-11-27
> > >
> > > Thank you so much for the endorsement!
> > >
> > > We have uploaded a new version with more results and improvements based on the valuable suggestions from the reviewers.
> > >
> > > Here are the results that we have added:
> > > - Communication overhead analysis and comparisons between asynchronous and synchronous settings in Appendix A.3.
> > > - Reward performances with long runs in Appendix A.3.
> > > - More details in the proof in Appendix B, e.g., the derivation of equation 34 with details, and the explanation of equation 36.
> > > - Time complexity analysis in Appendix A.1.
> > >
> > > Highlight: We are the first work to analyze asynchronous policy gradient methods with general function parameterization, and we achieve the SOTA sample complexity for global convergence.
> > >
> > > We hope this would help with the evaluation.

---

> > > > ### Comment · Reviewer_oswF · 2024-11-30
> > > > **Further reply to authors**
> > > >
> > > > Thanks to the authors for their further improvement of the paper. In the current version, the numerical section and the proof are in a better state. Figure 7 clearly highlights the advantage of the asynchronous method when applied to agents with heterogeneous efficiency. Thus, I will further increase my rating accordingly.

---

### Official Review · Reviewer_ui29 · 2024-11-03

**Soundness:** 3
**Presentation:** 3
**Contribution:** 3
**Rating:** 6
**Confidence:** 3

**Summary:**

This paper proposes a policy-based federated RL with an asynchronous setting to handle varying arrival times of policy gradient updates. Specifically, the authors analyzed the global and FOSP sample complexity as well as time complexity with a concrete algorithm design. The authors also provided simulation results on MuJoCo, which tackle sample and time complexity issues separately. The proposed method is more practical and can be adaptable to various computing heterogeneity scenarios.

**Strengths:**

* Numerical experiments on MuJoCo demonstrate impressive results that support the better time complexity of the proposed method
* Both FOSP and global sample complexity match the state-of-the-art while the global time complexity can have a tighter bound with heterogeneous arrival times

**Weaknesses:**

* The ultimate goal of federated RL is to find the trade-off between sample and communication complexity while the emphasis of this work on communication complexity/strategy is limited and not clear to me. Please elaborate more about what the threshold or event triggered for any agent to have the synchronization/communication with the server in your proposed framework.
* There are some typos in the manuscript. For example, you write *MoJuCo* instead of *MuJoCo* in the caption of Figures 3 and 4.

**Questions:**

* In Line 268, you mention *the set of active agents*. Does it mean the agents that can apply global iteration? If so, then the following paragraph mentions that *only one gradient to update the model from the agent who has finished its local computation.* In other words, does it allow more than one agent to apply policy gradient at the same iteration?
* For Figure 3, could you please let PG (N=1)  and AfedPG (N=2) train even longer to see if they can converge to a similar reward as the other two? If they cannot, I feel curious as to why they can't.
* Is there any analysis or experiment of communication cost?

---

> ### Author Response · Authors · 2024-11-24
>
> **W1.**
> Thank you so much for this suggestion! The communication overhead was not the target of this paper. However, after we accepted your valuable advice and dived deep into the analysis, we found that AFedPG actually has advantages compared to FedPG. We added communication overhead results in Appendix A.3, and answered Q3. We hope this could address your concerns.
>
> ---
>
> **W2.**
> Thank you! We have fixed the typos with another round of proofreading.
>
> ---
>
> **Q1.**
> In the asynchronous setting, concurrency roughly means the number of machines that are computing. In the synchronous setting, the server waits until it receives policy gradients from all $N$ agents, while in the asynchronous setting, the server runs as soon as one policy gradient is received. In a continuous space (time), it has measure zero that two discrete time spots are the same. Thus, only one agent has the possibility to be applied on the server, while the others keep computing.
>
> ---
>
> **Q2.**
> It takes a very long time to wait for them to converge to the optimal point, especially when the state and action spaces are large. We only use these experiments to show the benefit of more agents. To answer the question and verify this hypothesis, we test the result on the Swimmer-v4 task. They can converge to a similar reward, while the variance (shadow area) is larger. We assume the federated setting has the potential ability to reduce the variance when more agents engage, which could be an interesting direction for future study.
>
> ---
>
> **Q3.**
> Thank you for the suggestion! We have added the analysis of experiments in Appendix A.3 to further enhance it. We briefly state the key points here.
>
> In theory, the synchronous setting and asynchronous settings have the same cumulative communication cost as the sample complexities are the same, and each agent collects the same amount of samples in each update. Our analysis shows that the AFedPG could keep the SOTA sample complexity in policy gradient methods in RL in Table 1. Moreover, when we analyze the training process through the lens of fine-grained time, AFedPG shows several advantages.
>
> On the server side, in FedPG, it is noticeable that the communication mainly happens in a short time window in the downlink process, which brings a heavy burden with a peak, especially in the resource constraint scenario. In AFedPG, the communication is more evenly distributed, which does not have a peak burden and has less requirement for communication resources.
>
> On the agent side, in FedPG, all agents have the same communication burden. In AFedPG, the leader agent (fast) has more communication burden $\mathcal{O}(t_{\max})$, while the straggler agent (slow) has less communication burden $\mathcal{O}(t_{\min})$. This is a more rational allocation compared to the equality setting regardless of the heterogeneous computing resources. In practice, agents have heterogeneous resources, while FedPG does not consider this and evenly distributes the burden. The equal allocation could bring too much burden for the slow ones. In AFedPG, it shifts the burden to the fast ones naturally.
>
> We hope this section could address your concerns.

---

> > ### Comment · Reviewer_ui29 · 2024-11-26
> > **Reviewer Response**
> >
> > Thanks the authors for answering all my questions and further proofreading. However, I am still not quite satisfied with Q2, especially I assume that PG (N=1) is just a conventional approach that you may at least get the convergence eventually, which should already be shown in prior practical works. The authors mentioned that they tested the result on the Swimmer-v4 task and they can converge to a similar reward for PG (N=1) and AfedPG (N=2). Could you please point me out if you have updated in your manuscript? Also, it seems that it is especially hard to tell if Humanoid-v4 can converge to the optimal point in Figure 3 for PG(N=1) due to the fact that it is already gradually converging.

---

> > > ### Author Response · Authors · 2024-11-27
> > >
> > > We sincerely apologize for the confusion, and the claim before showing results. We did not put the partial results in the previous version. We have uploaded a new version, which includes the complete results in Appendix A.3.
> > >
> > > In Appendix A.3, it contains descriptions with details. We briefly state here. For Swimmer-v4, Hopper-v4, and Walker2D-v4 tasks, PG achieves similar rewards with AFedPG in Figure 9. For Humanoid-v4, in some runs, PG achieves the optimal rewards (shadowed area), while in some runs, PG does not achieve the reward compared to our AFedPG, because of the hyperparameter tuning of PG, e.g., learning rates.
> > >
> > > The hyperparameter tuning of PG has no contribution to our main claim. Recall that we aim to use these experiments to verify the speedup effect in AFedPG in Table 1. The suboptimal hyperparameters of PG in the Humanoid-v4 task do not influence the conclusion: The more agents in AFedPG, the faster the optimal reward will be achieved.
> > >
> > > ---
> > >
> > > Compared to the original version, we have added
> > > - Communication overhead analysis in Appendix A.3.
> > > - Reward performances with long runs in Appendix A.3.
> > > - More details in the proof in Appendix B, e.g., the derivation of equation 34 with details, and the explanation of equation 36.
> > > - Time complexity analysis in Appendix A.1.
> > >
> > > On the other hand, we are the first work to analyze asynchronous policy gradient methods with general function parameterization, and we achieve the SOTA sample complexity for global convergence.
> > >
> > > We hope this would help to address the main concerns. We appreciate a re-evaluation based on the improvement under your suggestions.

---

> > > > ### Comment · Reviewer_ui29 · 2024-11-28
> > > > **Reviewer esponse**
> > > >
> > > > Thanks to the authors for further experiments. I think the results now look reasonable to me. Due to the fact that the authors address mostly my concerns, I would like to increase score from 5 to 6.

---

### Official Review · Reviewer_KckT · 2024-11-04

**Soundness:** 3
**Presentation:** 3
**Contribution:** 2
**Rating:** 5
**Confidence:** 3

**Summary:**

This work investigates federated reinforcement learning with asynchronous synchronizations to improve the time complexity. They introduce the asynchronous federated policy gradient (AFedPG), which tackles lagged policies using a delay-adaptive lookahead. In addition, they present a sample complexity analysis of the algorithm, demonstrating a linear speedup compared to the single-agent scenario.

**Strengths:**

1. The work provides asynchronous synchronization updates tailored for federated RL.
2. The work presents a tight sample complexity analysis of the proposed algorithm, demonstrating a linear speedup that aligns with the single-agent state-of-the-art.

**Weaknesses:**

1. The application of asynchronous updates from federated learning to federated policy gradients appears to be incremental, especially since much of the supervised federated learning literature has examined how to manage lagged models, while existing federated reinforcement learning research focuses on addressing the dynamic nature of reinforcement learning in federated settings.
2. It appears that a momentum method was introduced for federated policy gradients in heterogeneous environments to handle online sample collections dependent on $\theta$ in [1]. While the paper emphasizes its novelty by discussing the momentum design (delay-adaptive lookahead), which differs from asynchronous supervised federated learning, it remains uncertain whether this concept is genuinely unique in comparison to prior literature in federated reinforcement learning, which also addresses the issue of online sample collections that vary with policy updates.

[1] Momentum for the Win: Collaborative Federated Reinforcement Learning across Heterogeneous Environments, Wang et al., ICML 2024

**Questions:**

.

---

> ### Author Response · Authors · 2024-11-24
>
> Thank you for bringing the valuable work in the synchronous setting! We hope that we have addressed your concerns. If there are any other questions, we would be happy to answer them.
>
> ---
>
> **W1.**
> We are the first work to analyze federated policy gradient methods with **General Policy Parameterization**, e.g., neural networks (Deep RL), for a **Global** Convergence, and achieve the SOTA results.
>
> Even without the asynchronous setting, there is no previous FedPG work that achieves a global convergence with general policy parameterization with the SOTA rates.
>
> Beyond that, we design the asynchronous method and further improve the performances, which is the main improvement and contribution in this paper.
>
> We also list 4 points in W2, which makes the novelty more clear and easy to understand. If there are any other concerns, we would be happy to answer them.
>
> ---
>
> **W2.**
> Thank you for bringing this synchronous work! It is indeed valuable and has several contributions to theoretical analysis. We have cited it in the updated version as it improves the results in FedRL. However, except in the synchronous setting, there are several main differences in terms of completeness and methodology:
>
> 1. We achieve a Global Convergence with the SOTA rate, while that work (FedM for short) only focuses on stationary point convergence. It seems that FedM cannot go through the global convergence analysis without new techniques.
>
> 2. We have General Function Parameterization, e.g., neural networks (Deep RL), results in both theory and experiments. That paper does not analyze the influence of general function parametrization, e.g., Deep RL, in theory. The general function approximation might bring higher sample complexity, especially for the global convergence.
>
> 3. In their setting, the local policies will not be in different steps (Fixed to the hyperparameter $K$ in their paper). In our asynchronous setting, we do not require that each agent is in the same step, which brings the second-order error term in equation 31 and unbounded variance. Thus, we design the lookahead mechanism to cancel the second-order errors, normalized updates to bound the variance, and thus, secure the convergence. This technique is new and unique.
>
> 4. Their momentum design is NOT the delay-adaptive lookahead method in our paper.
>
>     - The momentum is used for gradient estimation (not sampling) at Step 5 in Algorithm 1. We did not state this momentum part is a novelty in our method. Our delay-adaptive lookahead method is at Step 8 in Algorithm 1 for trajectory ''sampling'' at Step 2 in Algorithm 2. This is new and has never been used before in any FedRL works.
>
>     - On the other hand, the orders are different. In momentum methods, the momented point is ''between'' the old ($k-1$)-th and the new $k$-th points. In our delay-adaptive lookahead method, the new point $k$-th is ''between'' the old ($k-1$)-th and the momented point. This is the reason that we call it ''lookahead''.
>
>     - Roughly speaking, we have two momentums, but unlike the definition of the momentum gradient descent, the sampling one is not a conventional momentum.

---

> > ### Comment · Reviewer_KckT · 2024-11-26
> >
> > I appreciate the authors' clarifications. After carefully reading their responses and the other reviews, I remain unconvinced about the paper's contribution. I will maintain my score.

---

> > > ### Author Response · Authors · 2024-12-02
> > >
> > > Dear Reviewer KckT,
> > >
> > > We truly appreciate the responses. Here are the new results that we have added in the updated version as suggested:
> > >
> > > - Re-state the algorithms with clear explanations.
> > >
> > > - Time complexity analysis and explanations in Appendix A.1.
> > >
> > > - More explanations with details in the proof in Appendix B.
> > >
> > > - Reward performances with long runs in Appendix A.3.
> > >
> > > - Communication overhead analysis and comparisons between asynchronous and synchronous settings in Appendix A.3.
> > >
> > > We would like to briefly state our key contributions here. We are the first work to analyze the policy gradient methods in FedRL with general function parameterization and achieve the SOTA global convergence rate. The proposed delay-adaptive lookahead technique mitigates second-order errors in practical settings with lagged gradients, enabling guaranteed convergence in asynchronous FedRL -- a challenge not addressed in existing works. We improve the performances in communication cost and time consumption in an asynchronous setting, which is new.
> > >
> > > Do you have any specific aspects that we can clarify and further improve? Thank you so much for the suggestion!

---

### Official Review · Reviewer_T8LV · 2024-11-04

**Soundness:** 3
**Presentation:** 3
**Contribution:** 2
**Rating:** 5
**Confidence:** 4

**Summary:**

This paper proposes an asynchronous federated reinforcement learning framework. Then it introduces a delay-adaptive lookahead technique and employs normalized updates to integrate policy gradients to deal with the challenges brought by the asynchrony. Furthermore, the paper provides the theoretical global convergence bound. The experiments verify the improved performance of the proposed algorithm.

**Strengths:**

1. Convergence results are provided.
2. Asynchronous federated reinforcement learning framework is proposed.

**Weaknesses:**

1. This paper is not built on a federated framework. FedRL is designed to address heterogeneous environments and allow local agents to perform multiple iterations [1,2]. However, these are not considered in this paper.

[1] Momentum for the Win: Collaborative Federated Reinforcement Learning across Heterogeneous Environments, ICML24.
[2] Federated Reinforcement Learning with Environment Heterogeneity, AISTATS22.

2. This work lack necessary comparisons with current works. Actor-critic is a policy-based approach. This paper needs careful comparisons in details with [3] since both emphasize the asynchrony, not mentioned in Introduction briefly.
[3] Towards understanding asynchronous advantage actor-critic: convergence and linear speedup.

3. Technical contributions are limited. Authors claimed that even if all agents have an identical environment, each agent collects samples according to different policies because of the delay. This dynamic nature makes both the problem itself and the theoretical analysis challenging. However, this is somehow solved by [3]. The challenges brought by the features of Fed RL are not considered in this paper.

**Questions:**

4. Why does Proof of theorems lack the index of agent i? Since the server does not aggregate gradients or parameters from agents periodically, Fed RL is not applicable in this paper. Besides, it is just similar to [3]. Notations also make confusing.

5. What’s the technical contributions beyond existing FedRL? Technical differences of AFedPG compared to FedPG seems limited.

6. Authors first get the results of global convergence, then FOSP results. Why FOSP results are placed first in main text?

---

> ### Author Response · Authors · 2024-11-24
>
> Thank you for the suggestions in general. We sincerely apologize for the confusion. If there are any other questions, we would be happy to answer them.
>
> ---
>
> **W1.**
> The average among local updates is a senario in the conventional synchronous setting. In the asynchronous setting, the agent sends the update to the server as soon as it finishes the computation. In most conventional federated learning works, only the synchronous setting is considered. We aim to extend the scope to the asynchronous setting.
>
> ---
>
> **W2.**
> Thanks for the suggestion! We admit that the **Actor-Critic** (AC) method in [3] (not PG) contains novel contributions in theoretical analysis.
>
> However, it only includes linear parametrization, which is a very weak and unpractical result (**See W3**). With a general function parametrization, e.g., neural networks (Deep RL), the SOTA result (even in the single-agent setting) of the AC method is $\mathcal{O}({\epsilon}^{-3})$ [4]. It is worse than the SOTA result in policy gradient, which is $\mathcal{O}({\epsilon}^{-2.5})$. We aim to compare with the best results, and thus, choose to analyze PG methods.
>
> ---
>
> **W3.**
>
> 1. [3] only has **Linear Parametrization** (Deep RL is not included.), which is a very weak result with limited meaning. With a General Function Parametrization, e.g., neural networks (Deep RL), there is no such a result. We consider a general and practical setting with a General Function Parametrization.
>     - Even in the single-agent setting (without federated agents), the SOTA result of the **AC method is $\mathcal{O}({\epsilon}^{-3})$ [4]** and the previous approach is $\mathcal{O}({\epsilon}^{-6})$ [5], which is still worse than our $\mathcal{O}({\epsilon}^{-2.5})$. In the federated setting, there is no result that achieves $\mathcal{O}({\epsilon}^{-3})$ for AC methods. Moreover, with a general function parameterization, we compare the performances of their A3C in Figure 4, which is much worse.
>
> 2. [3] relies on a strong and unpractical assumption, Assumption 2. It assumes that the largest **delay is bounded** by a constant $K_{0}$. However, in practice, the slowest agent may not communicate with the server, and thus, has an infinite delay. In our analysis, we do not require any boundary for the largest delay, because we only contain the average delay in the convergence rate, and the average delay is naturally bounded by the number of agents $N$ in Lemma B.10 (our corollary).
>
> 3. [3] is an AC method with **extra value networks**, which requires much more computation and memory cost compared to the pure policy gradient method. Thus, the fine-tuning of Gemini and GPT4 uses policy gradient methods **instead of AC** methods.
>
> ---
>
> **Q4.**
> We aim to train a global policy in FedRL. The theorems give the convergence rates for the policy model w.r.t. the number of global updates on the **server**. It has **no relationship** with the index of agents. We achieve a result that the global convergence rate is upper bounded by the average delay, and the average delay is bounded by the number of agents $N$. This is new in FedRL. If there are any other questions, we would be happy to answer them.
>
> ---
>
> **Q5.**
> We are the first work to analyze federated policy gradient methods with General Policy Parameterization, e.g., neural networks (Deep RL), for a Global Convergence, and achieve the SOTA results.
>
> Even without the asynchronous setting, there is NO previous FedPG work that achieves a Global Convergence with General Policy Parameterization (Deep RL) with the SOTA rates.
>
> Beyond that, we design the asynchronous method and further improve the performances, which is the main contribution in this paper. Without our lookahead techniques, there is no global convergence rate in AFedPG with a general parameterization setting, because the second-order errors are hard to bound. We find that the asynchronous method (with our techniques) is not just as good, but better.
>
> ---
>
> **Q6.**
> Thank you for the suggestion. We have changed the order in the main paper.
>
> ---
>
> [4] Closing the Gap: Achieving Global Convergence (Last Iterate) of Actor-Critic under Markovian Sampling with Neural Network Parametrization. ICML 2024.
>
> [5] Single-Timescale Actor-Critic Provably Finds Globally Optimal Policy. ICLR 2021.

---

> > ### Comment · Reviewer_T8LV · 2024-11-26
> >
> > Thank you for your detailed responses. I still have a few questions:
> >
> > 1. In [3], the policy is also parameterized using General Policy Parameterization (Deep RL), rather than a linear parameterization.
> >
> > 2. Regarding FL, [6] considers multiple local steps at local agents when considering asynchrony. Since FL aims to reduce communication costs, I still believe that this paper does not fundamentally operate within a federated framework.
> >
> > [6] Anarchic Federated Learning, ICML2022.
> >
> > Based on your response, I have increased my rating for the paper.

---

> > > ### Author Response · Authors · 2024-11-30
> > >
> > > We truly appreciate the re-evaluation.
> > >
> > > In [3], it has a general parameterization for FOSP only. For the global convergence (Theorem 5), the critic is Linear (not deep RL), and the policy is softmax (not general) for discrete and finite action and state spaces only, which should not be restricted in deep RL. In deep RL (AC methods), the results are analyzed in [4] and [5] in the single-agent setting.
> > >
> > > We hope this would answer the question about the settings in previous RL works.

---

### Official Review · Reviewer_YjHR · 2024-11-08

**Soundness:** 3
**Presentation:** 2
**Contribution:** 2
**Rating:** 6
**Confidence:** 3

**Summary:**

The paper aims to enhance the efficiency of federated reinforcement learning (FedRL) by introducing an asynchronous framework, AFedPG, which leverages policy gradient (PG) updates from multiple agents without requiring synchronized updates. This approach is designed to address issues related to delayed updates and computational heterogeneity, which are common challenges in federated setups, especially with varying agent speeds and capacities.

**Strengths:**

Contributions claimed in the paper include,

--Proposes a new asynchronous FedRL algorithm (AFedPG) tailored to policy gradient updates, using a delay-adaptive lookahead technique to manage lagging updates in asynchronous settings.

-- Provides theoretical convergence guarantees, including global and first-order stationary point convergence, for the asynchronous federated policy-based RL.

-- Achieves a linear speedup in sample complexity with an increasing number of agents, reducing the per-agent complexity from $O(\epsilon^{-2.5})$ to $O(\epsilon^{-2.5}/N)$. (However, the proof is unclear and it is hard to see how the authors can avoid a dependence on the delay in the sample complexity.)

  -- Improves time complexity over synchronous methods by reducing the dependency on the slowest agent’s computational time, with gains highlighted in scenarios of high computational heterogeneity.

-- Empirically validates AFedPG's performance in various MuJoCo environments, demonstrating faster convergence (time-wise) over synchronous FedPG and other baselines.

**Weaknesses:**

In general, the paper is not clearly written. I don't see how the authors were able to avoid a dependence on the delay in their sample complexity. Their current derivations for bounding the error term (from the delay) have many typos and are hard to follow. Specific concerns/questions of the paper include:

-- Step 4 in Algorithm 2 is confusing. Where does the local agent get $d_{k-1}$ from? Did the authors mean $d_{k-\delta_k}$ instead? If the authors meant $d_{k-1}$, the current algorithm descriptions do not mention how $d_{k-1}$ can be made available to agent $i$.

-- A major component of the proof is bounding the error term $e_k := d_{k-\delta_k} - \nabla J(\theta_k)$, which arises from the delay. Equation (30) in the appendix provides a derivation of how $e_k$ can be expressed (and subsequently bounded). However, there seems to be serious typos in equation (30). For instance, in the first line, I am not sure why a term $d_{\delta_{k-1}}$ appears, when $e_k$ is actually $d_{k-\delta_k} - \nabla J(\theta_k)$. This makes it difficult to follow the argument in this derivation, and there is also no explanation of the derivation, which might have made it easier to follow the argument flow. Given that this is a particularly important term to bound to derive either first-order or global convergence rates, the authors should make an effort to clarify and explain these derivations.

-- The current convergence bound seems to have no dependence on the delay in the network, which is $N$ in the worst-case (e.g. assuming cyclic update). This is somewhat confusing to me; intuitively, even with a delay-adaptive step size for the $\theta$ update, there should be some price to pay for a cyclic delay structure. My current understanding is that perhaps the authors were able to bypass the dependence on the delay by their handling of the gradient-bias term $e_k$ (caused by the delay). However, given that the current derivation of bounding $e_k$ is highly unclear (see my earlier point), it is not clear to me whether the result as currently stated actually holds. If it holds, the authors should make it a lot clearer how and why they are able to avoid the dependence on the delay, as this is a key part of their contribution.

-- The definition of the global time is unclear. The authors should make it more precise, and have a formal statement and proof of their current stated bound on the global time being $O(\frac{\bar{t}\epsilon^{-2.5}}{N})$, where $\bar{t} = \frac{1}{\sum_{i=1}^N \frac{1}{t_i}}$.

--On a related note, the definition of $t_i$ seems a little unclear to me, given that at different iterations, agent $i$ might require varying amounts of time (i.e. there shouldn't be a single time complexity $t_i$ for each agent $i$). The authors should make their definition of what they mean by $t_i$ more precise.

**Questions:**

See my questions from the previous section.

---

> ### Author Response · Authors · 2024-11-24
>
> Thank you for reading the paper carefully! Hopefully, we have answered all your questions. If there are any other concerns, we would be happy to answer them.
>
> ---
>
> **W1.**
> We are sorry about the confusion. In the original algorithms, the $k$ on the server and agent sides are not the same one. To avoid confusion, we have reconstructed the algorithm description (Algorithm 1 and 2), while keeping all notations unchanged in the theoretical analysis. In the new description, a simple plugin between the equations will lead to the theoretical results. We hope this could help.
>
> ---
>
> **W2.**
> Thank you so much for bringing this! It is indeed a typo. It should be $d_{k-1-\delta_{k-1}}$ instead of $d_{\delta_{k-1}}$. We have fixed the typo, and it should be consistent now.
>
> We have $e_{k} = d_{k - \delta_{k}} - \nabla J(\theta_{k})$ and $d_{k - \delta_{k}} = (1 - \alpha_{k - \delta_{k}}) d_{k-1 - \delta_{k-1}}$
>  $+ \alpha_{k - \delta_{k}}$
> $g(\tau_{k - \delta_{k}}, \theta_{k - \delta_{k}})$. (It is unable to show \widetilde in $g(\cdot)$ here.)
>
> Combining them, we have
> $e_{k} = (1 - \alpha_{k - \delta_{k}}) d_{k-1 - \delta_{k-1}} - \nabla J(\theta_{k}) + \alpha_{k - \delta_{k}}$ $g(\tau_{k - \delta_{k}}, \theta_{k - \delta_{k}})$.
> Then, we can have the result in equation 31, and construct the recursive form.
>
> ---
>
> **W3.**
> In the worst case, the largest delay could be infinite (not communicate at all), while the average delay is upper bounded by $N$.
>
> We have a very interesting result in AFedPG. We do not make any assumptions about the delay, and the largest delay is allowed to be infinite. If some agents do not communicate at all, we simply lose their computation resources. The convergence rate is upper bounded by the average delay $\bar{\delta}$, and the average delay is bounded by $N$ in Lemma B.10. In equation 37, the term with the average delay is much smaller than the dominated term $\mathcal{O}(K^{-2.5})$ ($K$ could reach an order of magnitude of six or even larger.), and thus, does not hurt the convergence rate.
>
> ---
>
> **W4.**
>  Thank you for the suggestion! We have added the explanation with detailed derivation in Appendix A.1. It should be very easy to understand.
>
> ---
>
> **W5.**
> As the agent has the same computation requirement in each iteration (The number of collected samples is the same.), we assume that the time complexity in each iteration is the same. Sorry about the confusion. We have added this statement in line 392.

---

> ### Author Response · Authors · 2024-11-30
>
> Dear Reviewer,
>
> We have uploaded a new version with more results and improvements based on your valuable suggestions.
>
> Here are the results that we have added, as suggested by you:
> - Re-state the algorithms with clear explanations.
> - Time complexity analysis and explanations in Appendix A.1.
> - More explanations with details in the proof in Appendix B.
>
> Some main results suggested by the others:
> - Reward performances with long runs in Appendix A.3.
> - Communication overhead analysis and comparisons between asynchronous and synchronous settings in Appendix A.3.
>
> We hope this would answer your questions and address your concerns. We would truly appreciate it if you could re-evaluate the work accordingly.

---

> > ### Comment · Reviewer_YjHR · 2024-12-03
> >
> > I thank the reviewer for their significant efforts in revising the paper. The comments have mostly addressed my initial concerns. I thus raise the score to 6.

---

### Public Comment · ~Flint_Xiaofeng_Fan1 · 2025-04-17
**Recommendation to include FedPG‑BR (NeurIPS 2021)**

We congratulate the authors on introducing AFedPG, a novel asynchronous federated RL algorithm with rigorous convergence guarantees under heterogeneous delays. We respectfully note that FedPG‑BR (NeurIPS 2021):

1. Was the first FedPG framework to prove a formal sample‑complexity bound in the settings similar to AFedPG,

2. Additionally provides Byzantine resilience, tolerating up to 50 % adversarial agents.

A brief citation and discussion of FedPG‑BR in the related work would further strengthen the paper’s scholarly context. We hope the authors will kindly consider including this reference to acknowledge its foundational role in federated policy gradients.

Cheers,
Flint

https://flint-xf-fan.github.io/

---

### Meta-Review · Area_Chair_Mro7 · 2024-12-21

**Metareview:**

This paper introduces AFedPG, an asynchronous framework for federated reinforcement learning (FedRL), which utilizes policy gradient (PG) updates from multiple agents without requiring synchronized updates. The approach is specifically designed to address challenges in federated setups, such as delayed updates and computational heterogeneity due to varying agent speeds and capacities. The authors provide theoretical results, including first-order and global convergence rates, and conduct simulation experiments on MuJoCo to demonstrate improvements in sample and time complexity.

The reviewers raised several common concerns, including the need for clarification of notation and presentation, the absence of a communication cost analysis, and the novelty of the techniques. In their rebuttal, the authors successfully addressed these concerns. They clarified misunderstandings and corrected typos, leading to improved reviewer scores. Additionally, the inclusion of communication cost comparisons in the experiments strengthened the paper’s relevance to the federated setting. Regarding technical novelty, the authors convincingly argued that there were no prior results on the convergence of federated policy gradients for general policy parameterizations. Furthermore, their theoretical results outperform those of single-agent settings by achieving a linear speedup. These contributions, combined with the clarification of previously ambiguous points, make the paper a valuable addition to the FedRL literature. Based on these factors, I recommend weak acceptance of this paper.

**Additional Comments On Reviewer Discussion:**

The reviewers raised several common concerns, including the need for clarification of notation and presentation, the absence of a communication cost analysis, and the novelty of the techniques. In their rebuttal, the authors successfully addressed these concerns.

---

### Decision · Program_Chairs · 2025-01-22

Accept (Poster)